# ReMasker: Imputing Tabular Data with Masked Autoencoding

**Tianyu Du**[1][*]   **Luca Melis**[2]   **Ting Wang**[3,4]
[1]Zhejiang University   [2]Meta   [3]Penn State   [4]Stony Brook University

## Abstract

We present ReMasker, a new method of imputing missing values in tabular data by extending the masked autoencoding framework. Compared with prior work, ReMasker is both *simple* – besides the missing values (*i.e.*, naturally masked), we randomly "re-mask" another set of values, optimize the autoencoder by reconstructing this re-masked set, and apply the trained model to predict the missing values; and *effective* – with extensive evaluation on benchmark datasets, we show that ReMasker performs on par with or outperforms state-of-the-art methods in terms of both imputation fidelity and utility under various missingness settings, while its performance advantage often increases with the ratio of missing data. We further explore theoretical justification for its effectiveness, showing that ReMasker tends to learn missingness-invariant representations of tabular data. Our findings indicate that masked modeling represents a promising direction for further research on tabular data imputation. (code available: `https://github.com/alps-lab/remasker`)

## 1    Introduction

Missing values are ubiquitous in real-world tabular data due to various reasons during data collection, processing, storage, or transmission. It is often desirable to know the most likely values of missing data before performing downstream tasks (*e.g.,* classification or synthesis). To this end, intensive research has been dedicated to developing imputation methods ("imputers") that estimate missing values based on observed data (Yoon et al., 2019; Jarrett et al., 2022; Kyono et al., 2021; Stekhoven & Buhlmann, 2012; Mattei & Frellsen, 2018). Yet, imputing missing values in tabular data with high fidelity and utility remains an open problem, due to challenges including the intricate correlation across different features, the variety of missingness scenarios, and the scarce amount of available data with respect to the number of missing values.

The state-of-the-art imputers can be categorized as either *discriminative* or *generative*. The discriminative imputers, such as MissForest (Stekhoven & Buhlmann, 2012), MICE (van Buuren & Groothuis-Oudshoorn, 2011), and MIRACLE (Kyono et al., 2021), impute missing values by modeling their conditional distributions on the basis of other values. In practice, these methods are often hindered by the requirement of specifying the proper functional forms of conditional distributions and adding the set of appropriate regularizers. The generative imputers, such as GAIN (Yoon et al., 2019), MIWAE (Mattei & Frellsen, 2018), GAMIN (Yoon & Sull, 2020), and HI-VAE (Nazabal et al., 2020), estimate the joint distributions of all the features by leveraging the capacity of deep generative models and impute missing values by querying the trained models. Empirically, GAN-based methods often require a large amount of training data and suffer the difficulties of adversarial training (Goodfellow et al., 2014), while VAE-based methods often face the limitations of training through variational bounds (Zhao et al., 2022). Further, some of these methods either require complete data during training or operate on the assumptions of specific missingness patterns.

In this paper, we present ReMasker, a novel method that extends the masked autoencoding (MAE) framework (Devlin et al., 2018; He et al., 2022) to imputing missing values of tabular data. The idea of ReMasker is simple: Besides the missing values in the given dataset (*i.e.*, naturally masked), we randomly select and "re-mask" another set of values, optimize the autoencoder with the objective of

---

[*]The work was done when the first author was a postdoc at Penn State.

reconstructing this re-masked set, and then apply the trained autoencoder to predict the missing values. Compared with the prior work, REMASKER enjoys the following desiderata: (*i*) it is instantiated with Transformer (Vaswani et al., 2017) as its backbone, of which the self-attention mechanism is able to capture the intricate inter-feature correlation (Huang et al., 2020); (*ii*) without specific assumptions about the missingness mechanisms, it is applicable to various scenarios even if complete data is unavailable; and (*iii*) as the re-masking approach naturally accounts for missing values and encourages learning high-level representations beyond low-level statistics, REMASKER works effectively even under a high ratio of missing data (*e.g.*, up to 70%).

With extensive evaluation on 12 benchmark datasets under various missingness scenarios, we show that REMASKER performs on par with or outperforms 13 popular methods in terms of both imputation fidelity and utility, while its performance advantage often increases with the ratio of missing data. We further explore the theoretical explanation for its effectiveness. We find that REMASKER encourages learning *missingness-invariant* representations of tabular data, which are insensitive to missing values. Our findings indicate that, besides its success in the language and vision domains, masked modeling also represents a promising direction for future research on tabular data imputation.

## 2    RELATED WORK

**Tabular data imputation.** The existing imputation methods can be categorized as either discriminative or generative. The discriminative methods (Stekhoven & Buhlmann, 2012; van Buuren & Groothuis-Oudshoorn, 2011; Kyono et al., 2021) often specify a univariable model for each feature conditional on all others and perform cyclic regression over each target variable until convergence. Recent work has also explored adaptively selecting and configuring multiple discriminative imputers (Jarrett et al., 2022). The generative methods either implicitly train imputers as generators within the GAN framework (Yoon et al., 2019; Yoon & Sull, 2020) or explicitly train deep latent-variable models to approximate the joint distributions of all features (Mattei & Frellsen, 2018; Nazabal et al., 2020). There are also imputers based on representative-value (*e.g.*, mean) substitution (Hawthorne & Elliott, 2005), EM optimization (García-Laencina et al., 2010), matrix completion (Hastie et al., 2015), or optimal transport (Muzellec et al., 2020). The work is also related to that models missing data by adapting existing model architectures (Przewięźlikowski et al., 2021).

**Transformer.** Transformer has emerged as a dominating design (Vaswani et al., 2017) in the language domain, in which multi-head self-attention and MLP layers are stacked to capture both short- and long-term correlations between words. Recent work has explored the use of Transformer in the vision domain by treating each image as a grid of visual words (Dosovitskiy et al., 2020). For instance, it has been integrated into image generation models (Jiang et al., 2021; Zhang et al., 2021; Hudson & Zitnick, 2021), achieving performance comparable to CNN-based models. Recent work (Somepalli et al., 2021; Huang et al., 2020; Arik & Pfister, 2021) also explores applying Transformer to model tabular data, showing its promising expressive power.

**Masked autoencoding (MAE).** Autoencoding is a classical sel-supervised approach for learning representation (Vincent et al., 2008; Pathak et al., 2016): an encoder maps an input to its representation and a decoder reconstructs the original input. Meanwhile, masked modeling is originally proposed as a pre-training method in the language domain: by holding out a proportion of a word sequence, it trains the model to predict the masked tokens (Devlin et al., 2018; Radford & Narasimhan, 2018). Recent work has combined autoencoding and masked modeling in vision tasks (Dosovitskiy et al., 2020; Bao et al., 2022). The seminal MAE (He et al., 2022) represents the state of the art in self-supervised pre-training on the ImageNet-1K benchmark. In the tabular data domain, MET (Majmundar et al., 2022) employs the MAE approach to model tabular data. However, given its primary focus on representation learning, MET assumes data completeness and directly applies MAE.

To our best knowledge, this represents the first work to explore an extended MAE approach (with Transformer as the backbone) in the task of tabular data imputation.

## 3    REMASKER

Next, we present REMASKER, an extremely simple yet effective method for imputing missing values of tabular data. We begin by formalizing the imputation problem.

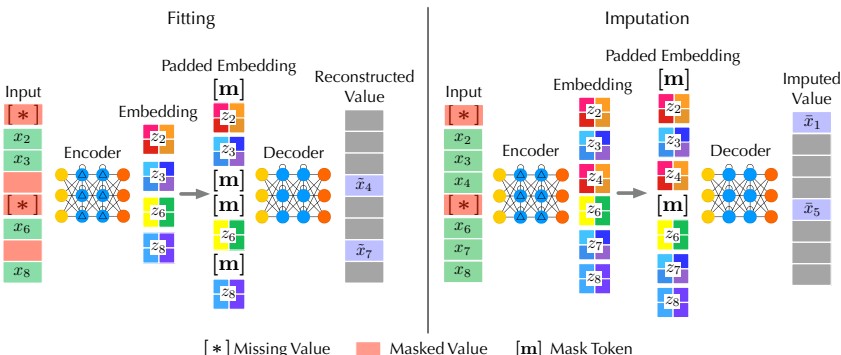

Figure 1: Overall framework of REMASKER. During the fitting stage, for each input, in addition to its missing values, another subset of values (re-masked values) is randomly selected and masked out. The encoder is applied to the remaining values to generate its embedding, which is padded with mask tokens and processed by the decoder to re-construct the re-masked values. During the imputation stage, the optimized model is applied to predict the missing values.

## 3.1 PROBLEM FORMALIZATION

**Incomplete data.** To model tabular data with $d$ features, we consider a $d$-dimensional random variable $\mathbf{x} \triangleq (\mathrm{x}_1, \ldots, \mathrm{x}_d) \in \mathcal{X}_1 \times \ldots \times \mathcal{X}_d$, where $\mathcal{X}_i$ is either continuous or categorical for $i \in \{1, \ldots, d\}$. The observational access to $\mathbf{x}$ is mediated by an mask variable $\mathbf{m} \triangleq (\mathrm{m}_1, \ldots, \mathrm{m}_d) \in \{0, 1\}^d$, which indicates the missing values of $\mathbf{x}$, such that $\mathrm{x}_i$ is accessible only if $\mathrm{m}_i = 1$. In other words, we observe $\mathbf{x}$ in its incomplete form $\tilde{\mathbf{x}} \triangleq (\tilde{\mathrm{x}}_1, \ldots, \tilde{\mathrm{x}}_d)$ with

$$\tilde{\mathrm{x}}_i \triangleq \begin{cases} \mathrm{x}_i & \text{if } \mathrm{m}_i = 1 \\ * & \text{if } \mathrm{m}_i = 0 \end{cases} \qquad (i \in \{1, \ldots, d\}) \tag{1}$$

where $*$ denotes the unobserved value.

**Missingness mechanisms.** Missing values occur due to various reasons. To simulate different scenarios, following the prior work (Yoon et al., 2019; Jarrett et al., 2022), we consider three missingness mechanisms: MCAR ("missing completely at random") – the missingness does not depend on the data, which indicates that $\forall \mathbf{m}, \mathbf{x}, \mathbf{x}', p(\mathbf{m}|\mathbf{x}) = p(\mathbf{m}|\mathbf{x}')$; MAR ("missing at random") – the missingness depends on the observed values, which indicates that $\forall \mathbf{m}, \mathbf{x}, \mathbf{x}'$, if the observed values of $\mathbf{x}$ and $\mathbf{x}'$ are the same, then $p(\mathbf{m}|\mathbf{x}) = p(\mathbf{m}|\mathbf{x}')$; and MNAR ("missing not at random") – the missingness depends on the missing values as well, which is the case if the definitions of MCAR and MAR do not hold. In general, it is impossible to identify the missingness distribution of MNAR without domain-specific assumptions or constraints (Ma & Zhang, 2021).

**Imputation task.** In this task, we are given an incomplete dataset $\mathcal{D} \triangleq \{(\tilde{\mathbf{x}}^{(i)}, \mathbf{m}^{(i)})\}_{i=1}^n$,[1] which consists of $n$ i.i.d. realizations of $\tilde{\mathbf{x}}$ and $\mathbf{m}$. The goal is to recover the missing values of each input $\tilde{\mathbf{x}}$ by generating an imputed version $\hat{\mathbf{x}} \triangleq (\hat{\mathrm{x}}_1, \ldots, \hat{\mathrm{x}}_d)$ such that

$$\hat{\mathrm{x}}_i \triangleq \begin{cases} \tilde{\mathrm{x}}_i & \text{if } \mathrm{m}_i = 1 \\ \bar{\mathrm{x}}_i & \text{if } \mathrm{m}_i = 0 \end{cases} \qquad (i \in \{1, \ldots, d\}) \tag{2}$$

where $\bar{\mathrm{x}}_i$ is the imputed value. Note that the imputation task is not concerned about optimizing data for concrete downstream tasks (e.g., training regression or generative models); such settings motivate concerns fundamentally entangled with each downstream task and often require optimizing the end objectives. Here, we focus solely on the imputation task itself.

## 3.2 DESIGN OF REMASKER

The REMASKER imputer extends the MAE framework (Dosovitskiy et al., 2020; Bao et al., 2022; He et al., 2022) that reconstructs masked components based on observed components. As illustrated in Figure 1, REMASKER comprises an encoder that maps the observed values to their representations and a decoder that reconstructs the masked values from the latent representations. However, unlike conventional MAE, as the data in the imputation task is inherently incomplete (*i.e.*, naturally masked),

---

[1] Without ambiguity, we omit the superscript $i$ in the following notations.

we employ a "re-masking" approach that explicitly accounts for this incompleteness in applying masking and reconstruction. At a high level, REMASKER works in two phases: *fitting* – it optimizes the model with respect to the given dataset, and *imputation* – it applies the trained model to predict the missing values of the dataset.

**Re-masking.** In the fitting phase, for each input $\tilde{\mathbf{x}}$, in addition to its missing values, we also randomly select and mask out another subset (*e.g.*, 25%) of $\tilde{\mathbf{x}}$'s values. Formally, letting $\mathbf{m}$ be $\tilde{\mathbf{x}}$'s mask, we define another mask vector $\mathbf{m}' \in \{0, 1\}^d$, which is randomly sampled without replacement, following a uniform distribution. Apparently, $\mathbf{m}$ and $\mathbf{m}'$ entail three subsets:

$$\mathcal{I}_{\text{mask}} = \{i | \mathbf{m}_i = 0\}, \quad \mathcal{I}_{\text{remask}} = \{i | \mathbf{m}_i = 1 \wedge \mathbf{m}'_i = 0\}, \quad \mathcal{I}_{\text{unmask}} = \{i | \mathbf{m}_i = 1 \wedge \mathbf{m}'_i = 1\} \quad (3)$$

Let $\tilde{\mathbf{x}}_{\overline{\mathbf{m}}}$, $\tilde{\mathbf{x}}_{\mathbf{m} \wedge \overline{\mathbf{m}'}}$, and $\tilde{\mathbf{x}}_{\mathbf{m} \wedge \mathbf{m}'}$ respectively be the masked, re-masked, and unmasked values. With a sufficient number of re-masked values, in addition to the missing values, we create a challenging task that encourages the model to learn missingness-invariant representations (more details in § 5). In the imputation phase, we do not apply re-masking.

**Encoder.** The encoder embeds each value using an encoding function and processes the resulting embeddings through a sequence of Transformer blocks. In implementation, we apply linear encoding function to each value $x$: $\text{enc}(x) = \mathbf{w}x + \mathbf{b}$, where $\mathbf{w}$ and $\mathbf{b}$ are learnable parameters. We also add positional encoding to $x$'s embedding to force the model to memorize $x$'s position in the input (*e.g.*, the $k$-th feature): $\text{pe}(k, 2i) = \sin(k/10000^{2i/d})$, where $k$ and $i$ respectively denote $x$'s position in the input and the dimension of the embedding, and $d$ is the embedding width. The encoder is only applied to the observed values: in the fitting phase, it operates on the observed values after re-masking (*i.e.*, the unmasked set $\mathcal{I}_{\text{unmask}}$); in the imputation phase, it operates on the non-missing values (*i.e.*, the union of re-masked and unmasked sets $\mathcal{I}_{\text{unmask}} \cup \mathcal{I}_{\text{remask}}$), as illustrated in Figure 1.

**Decoder.** The REMASKER decoder is instantiated as a sequence of Transformer blocks followed by an MLP layer. Different from the encoder, the decoder operates on the embeddings of both observed and masked values. Following (Devlin et al., 2018; He et al., 2022), we use a shared, learnable mask token as the initial embedding of each masked value. The decoder first adds positional encoding to the embeddings of all the values (observed and masked), processes the embeddings through a sequence of Transformer blocks, and finally applies linear projection to map the embeddings to scalar values as the predictions. Similar to (He et al., 2022), we use an asymmetric design with a deep encoder and a shallow decoder (*e.g.*, 8 versus 4 blocks), which often suffices to re-construct the masked values. Conventional MAE focuses on representation learning and uses the decoder only in the training phase. In REMASKER, the decoder is required to re-construct the missing values and is thus used in both fitting and imputation phases.

**Reconstruction loss.** Recall that the REMASKER decoder predicts the value for each input feature. We define the reconstruction loss functions as the mean square error (MSE) between the reconstructed and original values on (*i*) the re-masked set $\mathcal{I}_{\text{remask}}$ and (*ii*) unmasked set $\mathcal{I}_{\text{unmask}}$. We empirically experiment with different reconstruction loss functions (*e.g.*, only the re-masked set or both re-masked and unmasked sets).

Putting everything together, Algorithm 1 sketches the implementation of REMASKER.

---

**Algorithm 1** REMASKER

**Input:** $\mathcal{D} = \{(\tilde{\mathbf{x}}^{(i)}, \mathbf{m}^{(i)})\}_{i=1}^n$: incomplete dataset; remask: re-masking function; $f_\theta$, $d_\vartheta$: encoder and decoder; max_epoch: training epochs; $\ell$: reconstruction loss
**Output:** $\hat{\mathcal{D}} = \{(\hat{\mathbf{x}}^{(i)})\}_{i=1}^n$: imputed dataset

1: **while** max_epoch is not reached **do**          ▷ // fitting phase
2:      **for** $(\tilde{\mathbf{x}}, \mathbf{m}) \in \mathcal{D}$ **do**
3:          $\tilde{\mathbf{x}}_{\mathbf{m} \wedge \overline{\mathbf{m}'}}, \tilde{\mathbf{x}}_{\mathbf{m} \wedge \mathbf{m}'} \leftarrow$ remask$(\tilde{\mathbf{x}}, \mathbf{m})$;          ▷ // remasking
4:          $\mathbf{z} \leftarrow f_\theta(\tilde{\mathbf{x}}_{\mathbf{m} \wedge \mathbf{m}'})$;          ▷ // encoding unmasked values
5:          pad $\mathbf{z}$ with mask tokens;
6:      **end for**
7:      update $\theta$, $\vartheta$ by $\nabla\ell(d_\vartheta(\{\mathbf{z}\}), \{\tilde{\mathbf{x}}_{\mathbf{m} \wedge \overline{\mathbf{m}'}}\})$;          ▷ // minimizing reconstruction loss
8: **end while**
9: **for** $(\tilde{\mathbf{x}}, \mathbf{m}) \in \mathcal{D}$ **do**          ▷ // imputation phase
10:      $\mathbf{z} \leftarrow f_\theta(\tilde{\mathbf{x}}_{\mathbf{m}})$;          ▷ // encoding observed values
11:      pad $\mathbf{z}$ with mask tokens
12:      $\bar{\mathbf{x}}_{\overline{\mathbf{m}}} \leftarrow d_\vartheta(\mathbf{z})$;          ▷ // predicting missing values
13:      $\hat{\mathbf{x}} \leftarrow \tilde{\mathbf{x}}_{\mathbf{m}} \cup \bar{\mathbf{x}}_{\overline{\mathbf{m}}}$;
14: **end for**
15: **return** $\hat{\mathcal{D}} = \{\hat{\mathbf{x}}\}$.

---

## 4 EVALUATION

We empirically evaluate REMASKER in various scenarios using benchmark datasets. Our experiments are designed to answer the following key questions: (*i*) *Does* REMASKER *work?* – We compare REMASKER with a variety of state-of-the-art imputers in terms of imputation quality. (*ii*) *How does it work?* – We conduct an ablation study to assess the contribution of each component of REMASKER to its performance. (*iii*) *What is the best way of using* REMASKER*?* – We explore the use of REMASKER as a standalone imputer as well as one component of an ensemble imputer to understand its best practice.

**Datasets.** For reproducibility and comparability, similar to the prior work (Yoon et al., 2019; Jarrett et al., 2022), we use 12 real-world datasets from the UCI Machine Learning repository (Dua & Graff, 2017) with their characteristics deferred to § A.1.

**Missing mechanisms.** Following prior work (Yoon et al., 2019; Jarrett et al., 2022; Mattei & Frellsen, 2018; Hastie et al., 2015), we mainly focus on the MCAR and MAR settings. In MCAR, the mask vector of each input is realized following a Bernoulli random variable with a fixed mean. In MAR, with a random subset of features fixed to be observable, the remaining features are masked using a logistic model. For completeness, we also simulate one particular MNAR setting in experiments: the input features of MAR are further masked following a Bernoulli random variable with a fixed mean. We use HyperImpute (Jarrett et al., 2022) to simulate the above missingness mechanisms.

**Baselines.** We compare REMASKER with 13 state-of-the-art imputation methods: HyperImpute (Jarrett et al., 2022), MIWAE (Mattei & Frellsen, 2018), EM (García-Laencina et al., 2010), GAIN (Yoon et al., 2019), ICE, MICE, MIRACLE (Kyono et al., 2021), MissForest (Stekhoven & Buhlmann, 2012), Mean (Hawthorne & Elliott, 2005), Median, Frequent, Sinkhorn (Muzellec et al., 2020), and SoftImpute (Hastie et al., 2015), with details deferred to § A.2

**Metrics.** For each imputation method, we evaluate the imputation fidelity and utility by comparing its imputed data with the ground-truth data. In terms of fidelity, we mainly use two metrics: root mean square error (RMSE) to measure how the individual imputed values match the ground-truth data, and the Wasserstein distance (WD) to measure how the imputed distribution matches the ground-truth distribution. In terms of utility, we use area under the receiver operating characteristic curve (AUROC) as the metric on applicable datasets (*i.e.*, ones associated with classification tasks). In the case of multi-class classification, we use the one versus rest (OvR) setting. To be fair, we use logistic regression as the predictive model across all the cases.

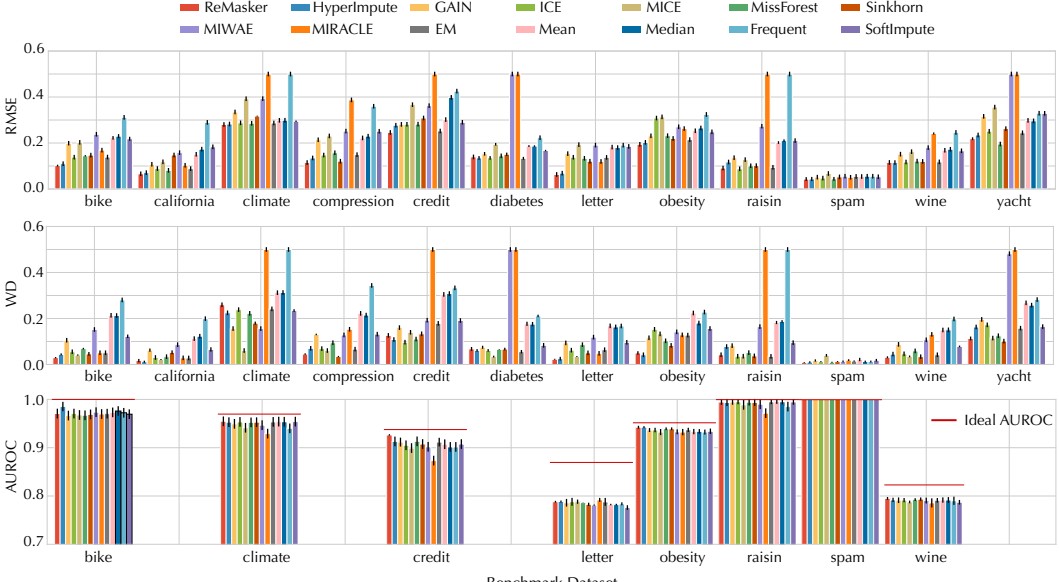

Figure 2: Overall performance of REMASKER and baseline imputers on 12 benchmark datasets under MAR with 0.3 missingness ratio. The results are shown as the mean and standard deviation of RMSE, WD, and AUROC scores (AUROC is only applicable to datasets with classification tasks). Note that REMASKER outperforms all the baseline imputers under at least one metric across all the datasets.

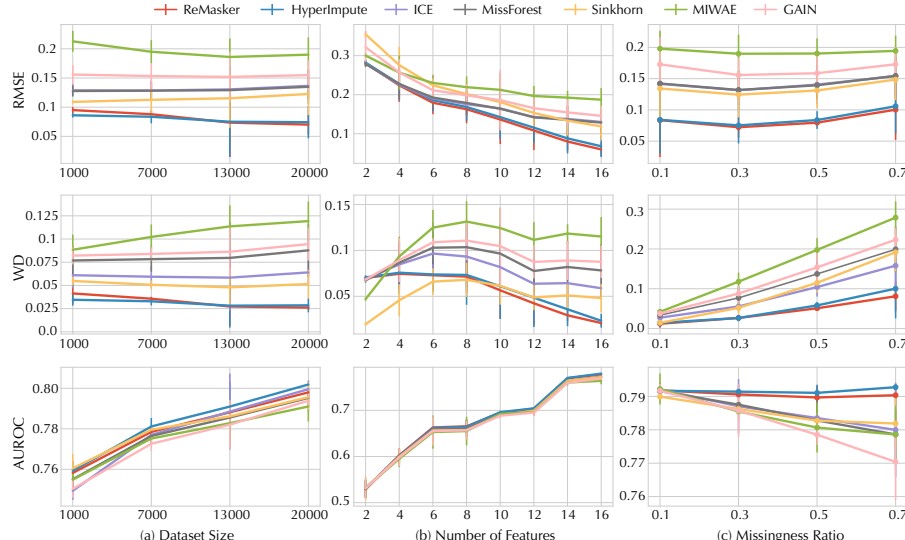

Figure 3: Sensitivity analysis of REMASKER on the `letter` dataset under the MAR setting. The scores are measured with respect to (a) the dataset size, (b) the number of features, and (c) the missingness ratio. The default setting is as follows: dataset size = 20,000, number of features = 16, and missingness ratio = 0.3.

### 4.1 OVERALL PERFORMANCE

We evaluate REMASKER and baseline imputers on the benchmark datasets under the MAR setting with 0.3 missingness ratio, with results summarized in Figure 2. Observe that REMASKER consistently outperforms all the baselines in terms of both fidelity (measured by RMSE and WD) and utility (measured by AUROC) across all the datasets. Recall that the benchmark datasets are collected from a variety of domains with highly varying characteristics (*cf.* Table 6): the dataset size varies from 308 to 20,060, while the number of features ranges from 7 to 57. Its superior performance across all the datasets demonstrates that REMASKER effectively models the intricate correlation among different features, even if the amount of available data is scarce. The only imputer with performance close to REMASKER is HyperImpute (Jarrett et al., 2022), which is an ensemble method that integrates multiple imputation models and automatically selects the most fitting model for each column of the given dataset. This highlights that the modeling capacity of REMASKER's masked autoencoder is comparable with ensemble models. In § B.4, we conduct a more comprehensive evaluation by simulating all three missingness scenarios (MCAR, MAR, and MNAR) with different missingness ratios. The results show that REMASKER consistently performs better across a range of settings.

### 4.2 SENSITIVITY ANALYSIS

To assess the factors influencing REMASKER's performance, we conduct sensitivity analysis by varying the dataset size, the number of features in the dataset, and the missingness ratio under the MAR setting. Figure 3 shows the performance of REMASKER within these experiments against the six closest competitors (HyperImpute, ICE, MissForest, GAIN, MIWAE, and Sinkhorn) on the `letter` dataset. We have the following observations. (*a*) The performance of REMASKER improves with the size of available data, while its advantage over other imputers (with the exception of HyperImpute) grows with the dataset size. (*b*) The number of features has a significant impact on the performance of REMASKER, with its advantage over other imputers increasing steadily with the number of features. This may be explained by that REMASKER relies on learning the holistic representations of inputs, while including more features contributes to better representation learning. (*c*) REMASKER is fairly insensitive to the missingness ratio. For instance, even with 0.7 missingness ratio, it achieves RMSE below 0.1, suggesting that it effectively fits sparse datasets. In § B.5, we also conduct an evaluation on other datasets with similar observations.

### 4.3 ABLATION STUDY

We conduct an ablation study of REMASKER to understand the contribution of different components to its performance using the `letter` dataset. Results on other datasets are deferred to § B.6.

| depth | RMSE | WD | AUROC |
|---|---|---|---|
| 2 | 0.0729 | 0.0263 | 0.7898 |
| 4 | 0.0636 | 0.0228 | 0.7903 |
| 6 | 0.0616 | 0.0219 | 0.7909 |
| 8 | 0.0611 | 0.0217 | 0.7892 |
| 10 | 0.0673 | 0.0245 | 0.7879 |

(a) Decoder depth

| width | RMSE | WD | AUROC |
|---|---|---|---|
| 16 | 0.0902 | 0.0379 | 0.7902 |
| 32 | 0.0714 | 0.0289 | 0.7885 |
| 64 | 0.0616 | 0.0219 | 0.7909 |
| 128 | 0.0795 | 0.0305 | 0.7845 |
| 256 | 0.1040 | 0.0403 | 0.7868 |

(b) Embedding width

| depth | RMSE | WD | AUROC |
|---|---|---|---|
| 2 | 0.0637 | 0.0239 | 0.7887 |
| 4 | 0.0625 | 0.0236 | 0.7877 |
| 6 | 0.0644 | 0.0239 | 0.7889 |
| 8 | 0.0616 | 0.0219 | 0.7909 |
| 10 | 0.0637 | 0.0227 | 0.7878 |

(c) Encoder depth

Table 1. Ablation study of REMASKER on the `letter` dataset. The default setting is as follows: encoder depth = 8, decoder depth = 6, embedding width = 64, masking ratio = 50%, and training epochs = 600.

| backbone | letter | | | california | |
|---|---|---|---|---|---|
| | RMSE | WD | AUROC | RMSE | WD |
| Transformer | 0.0611 | 0.0217 | 0.7892 | 0.0663 | 0.0172 |
| Linear | 0.1732 | 0.1604 | 0.7821 | 0.1786 | 0.1329 |
| Convolutional | 0.1694 | 0.1582 | 0.7836 | 0.1715 | 0.1286 |

Table 2. Performance with different backbones. (note: AUROC is inapplicable to the `california` dataset)

| loss | letter | | | california | |
|---|---|---|---|---|---|
| | RMSE | WD | AUROC | RMSE | WD |
| $\mathcal{I}_{\text{mask+}} \cup \mathcal{I}_{\text{unmask}}$ | 0.0616 | 0.0219 | 0.7909 | 0.0663 | 0.0172 |
| $\mathcal{I}_{\text{mask+}}$ | 0.0629 | 0.0237 | 0.7890 | 0.0840 | 0.0311 |
| $\mathcal{I}_{\text{unmask}}$ | 0.2079 | 0.1129 | 0.7901 | 0.1932 | 0.1906 |

Table 3. Performance of REMASKER with reconstruction loss w/ or w/o unmasked values.

**Model design.** The encoder and decoder of REMASKER can be flexibly designed. Here, we study the impact of three key parameters, the encoder depth (the number of Transformer blocks), the embedding width (the dimensionality of latent representations), and the decoder depth, with results summarized in Table 1a, Table 1b, and Table 1c. Observe that the performance of REMASKER reaches its peak with a proper model configuration (encoder depth = 8, decoder depth = 8, and embedding width = 64). This observation suggests that the model complexity needs to fit the given dataset: it needs to be sufficiently complex to effectively learn the holistic representations of inputs but not overly complex to overfit the dataset. We also compare the performance of REMASKER with different backbone models (*i.e.*, Transformer, linear, and convolutional) with the number of layers and the size of each layer fixed as the default setting. As shown in Table 2, Transformer-based REMASKER largely outperforms the other variants, which may be explained by that the self-attention mechanism can effectively capture the intricate inter-feature correlation under limited data Huang et al. (2020).

**Reconstruction loss.** We define the reconstruction loss as the error between the reconstructed and original values on the re-masked values $\mathcal{I}_{\text{mask+}}$ and the unmasked values $\mathcal{I}_{\text{unmask}}$. We measure the performance of REMASKER under three different settings of the construction loss: (*i*) $\mathcal{I}_{\text{mask+}} \cup \mathcal{I}_{\text{unmask}}$, (*ii*) $\mathcal{I}_{\text{mask+}}$ only, and (*iii*) $\mathcal{I}_{\text{unmask}}$ only on the `letter` and `california` datasets, with results shown in Table 3. Observe that using the reconstruction of unmasked values only is insufficient and yet including the reconstruction loss of unmasked values improves the performance, which is especially the case on the `california` dataset. This finding is different from the vision domain in which computing the loss on unmasked image patches reduces accuracy (He et al., 2022). We hypothesize that this difference is explained as follows. Unlike conventional MAE, due to the naturally missing values in tabular data, relying on re-masked values provides limited supervisory signals. Moreover, while images are signals with heavy spatial redundancy (*i.e.*, a missing patch can be recovered from its neighboring patches), tabular data tends to be highly semantic and information-dense. Thus, including the construction loss of unmasked values improves the model training.

## 4.4 PRACTICE OF REMASKER

**Training regime.** The ablation study above by default uses 600 training epochs. Figure 4(a) shows the impact of training epochs, in which we vary the training epochs from 100 to 1,600 and measure the performance of REMASKER on the `letter` dataset. Observe that the imputation performance

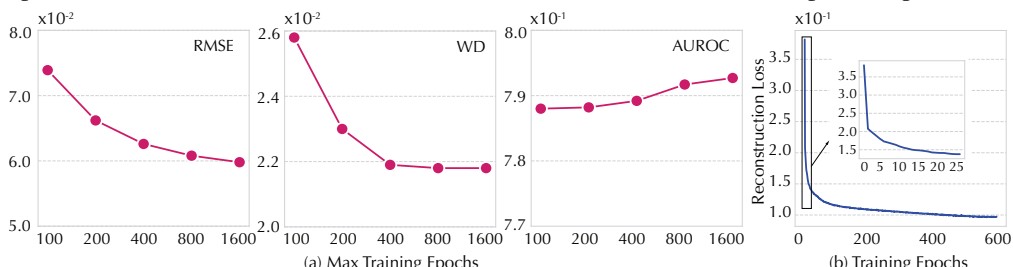

(a) Max Training Epochs

Figure 4: (a) REMASKER performance with respect to the maximum number of training epochs; (b) Convergence of REMASKER's reconstruction loss. Performed on `letter` under MAR with 0.3 missingness ratio.

| masking ratio | letter | | | california | |
|---|---|---|---|---|---|
| | RMSE | WD | AUROC | RMSE | WD |
| 0.1 | 0.0668 | 0.0215 | 0.0789 | 0.0888 | 0.0230 |
| 0.3 | 0.0562 | 0.0207 | 0.7897 | 0.0654 | 0.0151 |
| 0.5 | 0.0554 | 0.0212 | 0.7935 | 0.0663 | 0.0172 |
| 0.7 | 0.0906 | 0.0366 | 0.7878 | 0.1320 | 0.0650 |

Table 4. Performance with varying masking ratio. The results are evaluated on `letter` and `california` under MAR with 0.3 missingness ratio.

| base imputer | letter | | | california | |
|---|---|---|---|---|---|
| | RMSE | WD | AUROC | RMSE | WD |
| default | 0.0564 | 0.0215 | 0.7899 | 0.0722 | 0.0134 |
| REMASKER | 0.0554 | 0.0212 | 0.7935 | 0.0702 | 0.0115 |

Table 5. REMASKER as the base imputer within HyperImpute. The results are evaluated on `letter` and `california` under 0.3 MAR.

improves (as RMSE and WD decrease and AUROC increases) steadily with longer training and does not fully saturate even at 1,600 epochs. However, for efficient training, it is often acceptable to terminate earlier (*e.g.*, 600 epochs) with sufficient imputation performance. To further validate the trainability of REMASKER, with the maximum number of training epochs fixed at 600 (which affects the learning rate scheduler), we measure the reconstruction loss as a function of the training epochs. As shown in Figure 4(b), the loss quickly converges to a plateau within about 100 epochs and steadily decreases after that, demonstrating the trainability of REMASKER.

**Masking ratio.** The masking ratio controls the number of re-masked values (after excluding missing values). Table 4 shows its impact on REMASKER. Observe that the optimal ratio differs across different datasets, which may be explained by the varying number of features of different datasets (16 versus 9 in `letter` and `california`). Intuitively, a larger number of features affords a higher masking ratio to balance (*i*) encouraging the model to learn missingness-invariant representations and (*ii*) having sufficient supervisory signals to facilitate the training.

**Standalone vs. ensemble.** Besides using REMASKER as a standalone imputer, we explore its use as a base imputer within the ensemble imputation framework of HyperImpute, with results summarized in Table 5. It is observed that compared with the default setting (with mean substitution as the base imputer), using REMASKER as the base imputer improves the imputation performance, suggesting another effective way of operating REMASKER.

## 5 DISCUSSION

**Q1: What is REMASKER learning?** The empirical evaluation above shows REMASKER's superior performance in imputing missing values of tabular data. Next, we provide theoretical justification for its effectiveness. By extending the siamese form of MAE (Kong & Zhang, 2022), we show that REMASKER encourages learning *missingness-invariant* representations of input data, which requires a holistic understanding of the data even in the presence of missing values.

Let $f_\theta(\cdot)$ and $d_\vartheta(\cdot)$ respectively be the encoder and decoder. For given input $\mathbf{x}$, mask $\mathbf{m}$, and re-mask $\mathbf{m}'$, the reconstruction loss of REMASKER training is given by (here we focus on the reconstruction of re-masked values):

$$\ell(\mathbf{x}, \mathbf{m}, \mathbf{m}') = \|d_\vartheta(f_\theta(\mathbf{x} \odot \mathbf{m} \odot \mathbf{m}')) \odot (1 - \mathbf{m}') \odot \mathbf{m} - \mathbf{x} \odot (1 - \mathbf{m}') \odot \mathbf{m}\|^2 \quad (4)$$

where $\odot$ denotes element-wise multiplication. Let $\mathbf{m}^+ \triangleq \mathbf{m} \odot \mathbf{m}'$ and $\mathbf{m}^- \triangleq \mathbf{m} \odot (1 - \mathbf{m}')$. Eq (4) can be simplified as: $\ell(\mathbf{x}, \mathbf{m}^+, \mathbf{m}^-) = \|d_\vartheta(f_\theta(\mathbf{x} \odot \mathbf{m}^+)) \odot \mathbf{m}^- - \mathbf{x} \odot \mathbf{m}^-\|^2$. As the embedding dimensionality is typically much larger than the number of features, it is possible to make the autoencoder lossless. In other words, for a given encoder $f_\theta(\cdot)$, there exists a decoder $d_{\vartheta'}(\cdot)$, such that $d_{\vartheta'}(f_\theta(\mathbf{x} \odot \mathbf{m}^-)) \odot \mathbf{m}^- \approx \mathbf{x} \odot \mathbf{m}^-$. We can further re-write Eq (4) as:

$$\ell(\mathbf{x}, \mathbf{m}^+, \mathbf{m}^-) = \|d_\vartheta(f_\theta(\mathbf{x} \odot \mathbf{m}^+)) \odot \mathbf{m}^- - d_{\vartheta*}(f_\theta(\mathbf{x} \odot \mathbf{m}^-)) \odot \mathbf{m}^-\|^2$$
$$\text{s.t.} \quad \vartheta^* = \arg\min_{\vartheta'} \mathbb{E}_{\mathbf{x}'} \|d_{\vartheta'}(f_\theta(\mathbf{x}' \odot \mathbf{m}^-)) \odot \mathbf{m}^- - \mathbf{x}' \odot \mathbf{m}^-\|^2 \quad (5)$$

We define a new distance metric $\Delta_{\vartheta, \vartheta'}(\mathbf{z}, \mathbf{z}') \triangleq \|(d_\vartheta(\mathbf{z}) - d_{\vartheta'}(\mathbf{z}')) \odot \mathbf{m}^-\|^2$. Then, Eq (4) is reformulated as:

$$\ell(\mathbf{x}, \mathbf{m}^+, \mathbf{m}^-) = \Delta_{\vartheta, \vartheta*}(f_\theta(\mathbf{x} \odot \mathbf{m}^+), f_\theta(\mathbf{x} \odot \mathbf{m}^-))$$
$$\text{s.t.} \quad \vartheta^* = \arg\min_{\vartheta'} \mathbb{E}_{\mathbf{x}'} \|d_{\vartheta'}(f_\theta(\mathbf{x}' \odot \mathbf{m}^-)) \odot \mathbf{m}^- - \mathbf{x}' \odot \mathbf{m}^-\|^2 \quad (6)$$

Note that optimizing Eq (6) essentially minimizes the difference between $\mathbf{x}$'s representations under $\mathbf{m}^+$ and $\mathbf{m}^-$ (with respect to the decoder). As $\mathbf{m}^+$ and $\mathbf{m}^-$ mask out different values, this formulation promotes learning representations insensitive to missing values.

To validate the analysis above, we empirically measure the CKA similarity (Kornblith et al., 2019) between the latent representations (*i.e.*, the output of REMASKER's encoder) of complete inputs and inputs with missing values, with results shown in Figure 5. Observe that the CKA under different missingness ratios all steadily increase with the training length, indicating that REMASKER tends to learn missingness-invariant representations, which may explain its imputation effectiveness.

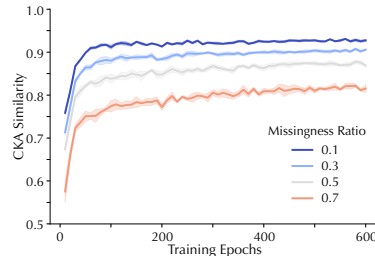

Figure 5: CKA similarity between the representations of complete and incomplete inputs (with the number of missing values controlled by the missingness ratio). The tested model is trained on `letter` under the MAR setting with 0.5 masking ratio.

Note that while the idea of masking observable values is also explored in prior work (Gondara & Wang, 2018), its adaptation in the context of MAE admits a more elegant imputation solution: it does not require initializing missing or re-masked values in the input while representing them with the same trainable tokens in the latent space encourages learning the distribution of re-masked values close to the distribution of missing values.

**Q2: How is REMASKER's performance influenced by the missingness mechanism?** As revealed above, REMASKER encourages learning representations invariant to re-masked values and then leverages such representations to impute missing values. Thus, while REMASKER takes effect as long as the distributions of re-masked and missing values are correlated, its performance may vary with the disparity between the two distributions. In MCAR, both re-masked and missing values follow similar distributions (*i.e.*, evenly distributed across different features). In MAR, as the observable features are fixed, the re-masked values are more likely to be selected from the observable features, which biases the representation learning toward the observable features. This bias is reflected in our experimental results, in which REMASKER tends to perform better under MCAR. In MNAR, as the missingness depends on the missing values themselves, it causes even more disparity between the distributions of re-masked and missing values, which also explains REMASKER's better performance under MAR and MCAR compared with MNAR.

**Q3: Why is Transformer effective for tabular data imputation?** While the theoretical understanding of the Transformer architecture is still limited, we borrow insights from recent work to explain the effectiveness of Transformer for tabular data imputation. Specifically, it is shown in (Park & Kim, 2022) that the multi-head self-attention (MSA) mechanism in Transformer essentially performs (optimizable) smoothing over the latent representations of different tokens, which makes Transformer robust against severe occlusions or perturbations (Naseer et al., 2021). In the same vein, in the context of tabular data imputation, MSA benefits learning representations that are invariant to missing values.

**Q4: What are REMASKER's limitations?** One kind of bias we observe is that in some cases (*e.g.*, the `climate` dataset in Figure 2), REMASKER outperforms alternative methods in terms of RMSE but underperforms in terms of WD. One explanation is that REMASKER is trained to optimize the reconstruction loss measured by MSE. Thus, it is biased towards re-constructing individual missing values rather than the distributions of missing values. Also, as REMASKER solely focuses on the imputation task itself, if the downstream tasks are known, its performance may not be optimal. In such contexts, it is essential to optimize REMASKER with respect to concrete tasks such as uncertainty estimation (van Buuren & Groothuis-Oudshoorn, 2011; Gondara & Wang, 2018).

## 6 CONCLUSION

In this paper, we conduct a pilot study exploring the masked autoencoding approach for tabular data imputation. We present REMASKER, a novel imputation method that learns missingness-invariant representations of tabular data and effectively imputes missing values under various scenarios. With extensive evaluation on benchmark datasets, we show that REMASKER outperforms state-of-the-art methods in terms of both imputation utility and fidelity. Our findings indicate that masked tabular modeling represents a promising direction for future research on tabular data imputation.

ACKNOWLEDGMENTS

The authors Ting Wang and Tianyu Du are partially supported by the National Science Foundation under Grant No. 1951729, 1953813, 2119331, and 2212323. Any opinions, findings, and conclusions or recommendations expressed in this material are those of the author(s) and do not necessarily reflect the views of the National Science Foundation.

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

# A  EXPERIMENTAL DETAILS

## A.1  DATASETS

Table 6 summarizes the characteristics of the datasets used in our experiments.

| Experiment Dataset | Dataset Size | # Features |
|---|---|---|
| (California) Housing | 20,640 | 9 |
| (Climate) Model Simulation Crashes | 540 | 18 |
| Concrete (Compressive) Strength | 1,030 | 9 |
| (Diabetes) | 442 | 10 |
| Estimation of (Obesity) Levels | 2,111 | 17 |
| (Credit) Approval | 690 | 15 |
| (Wine) Quality | 1,599 | 12 |
| (Raisin) | 900 | 8 |
| (Spam) Base | 4,601 | 57 |
| (Bike) Sharing Demand | 8,760 | 14 |
| (Letter) Recognition | 20,000 | 16 |
| (Yacht) Hydrodynamics | 308 | 7 |

Table 6. Characteristics of the datasets used in the experiments.

## A.2  BASELINES

We compare REMASKER with 13 state-of-the-art imputation methods: HyperImpute (Jarrett et al., 2022), a hybrid imputer that performs iterative imputation with automatic model selection; MI-WAE (Mattei & Frellsen, 2018), an autoencoder model that fits missing data by optimizing a variational bound; EM (García-Laencina et al., 2010), an iterative imputer based on expectation-maximization optimization; GAIN (Yoon et al., 2019), a generative adversarial imputation network that trains the discriminator to classify the generator's output in an element-wise manner; ICE, an iterative imputer based on regularized linear regression; MICE, an ICE-like, iterative imputer based on Bayesian ridge regression; MIRACLE (Kyono et al., 2021), an iterative imputer that refines the imputation of a baseline by simultaneously modeling the missingness generating mechanism; MissForest (Stekhoven & Buhlmann, 2012), an iterative imputer based on random forests; Mean (Hawthorne & Elliott, 2005), Median, and Frequent, which impute missing values using column-wise unconditional mean, median, and the most frequent values, respectively; Sinkhorn (Muzellec et al., 2020), an imputer trained through the optimal transport metrics of Sinkhorn divergences; and SoftImpute (Hastie et al., 2015), which performs imputation through soft-thresholded singular value decomposition.

## A.3  PARAMETER SETTING

The default parameter setting of REMASKER is listed in Table 7.

| model | parameter | setting |
|---|---|---|
| | optimizer | Adam |
| | initial learning rate | 1e-3 |
| | LR scheduler | cosine annealing |
| global | gradient clipping threshold | 5.0 |
| | training epochs | 600 |
| | batch size | 64 |
| | masking ratio | 0.5 |
| | Transformer block | 8 |
| encoder | embedding width | 64 |
| | number of heads | 4 |
| | Transformer block | 4 |
| decoder | embedding width | 64 |
| | number of heads | 4 |

Table 7. Default parameter setting of REMASKER.

# B  ADDITIONAL RESULTS

## B.1  EXECUTION EFFICIENCY

One design objective of MAE (*e.g.*, masking out over 75% of an image) is to scale up the training on large-scale image datasets without sacrificing its performance. This benefit is naturally inherited by REMASKER. We measure the running time (including both training and imputation) of REMASKER (under the default setting of Table 7) on datasets with over 1,000 records. For a comparison, we also measure the running time of HyperImpute under the same setting, with results summarized in Table 8.

| | compression | wine | spam | credit | bike | obesity | california |
|---|---|---|---|---|---|---|---|
| HyperImpute | 154.9s | 297.7s | 185.8s | 39.9s | 259.9s | 196.1s | 195.8s |
| REMASKER | 72.9s | 73.8s | 74.1s | 74.0s | 66.9s | 66.8s | 66.9s |

Table 8. Total running time of HyperImpute and REMASKER under the default setting.

Observe that compared with HyperImpute, REMASKER is not only more scalable but also less sensitive to data size.

## B.2  GENERALIZATION TO NEW DATA

To evaluate REMASKER's ability to generate new data, we run the following evaluation. We halve each dataset into two subsets $\mathcal{D}$ and $\mathcal{D}'$ (both with 30% missing value missing under MAR). We train REMASKER on $\mathcal{D}$ and then apply REMASKER to impute the missing values in $\mathcal{D}'$. The results are shown in Table 9. Note that compared with the setting where REMASKER is trained and applied on the same dataset (Figure 2), REMASKER performs comparably well under the setting that it is applied on a new dataset, indicating its generalizability.

| | compression | wine | spam | credit | bike | obesity | california |
|---|---|---|---|---|---|---|---|
| RMSE $\downarrow$ | 0.117 | 0.120 | 0.052 | 0.251 | 0.108 | 0.197 | 0.068 |
| WD $\downarrow$ | 0.047 | 0.033 | 0.009 | 0.131 | 0.035 | 0.059 | 0.020 |
| AUROC $\uparrow$ | N/A | 0.784 | 0.098 | 0.921 | 0.968 | 0.937 | N/A |

Table 9. Performance of REMASKER in imputing new datasets.

## B.3  CKA MEASURES OF INPUTS UNDER MASKS $\mathbf{m}^+$ AND $\mathbf{m}^-$

To further validate the analysis in § 5, we empirically measure the CKA similarity (Kornblith et al., 2019) between the latent representations (*i.e.*, the output of REMASKER's encoder) of inputs under the masks of $\mathbf{m}^+$ and $\mathbf{m}^-$, with results shown in Figure 6. Observe that while $\mathbf{m}^+$ and $\mathbf{m}^-$ represent different missing mechanisms, the CKA measures between the inputs under $\mathbf{m}^+$ and $\mathbf{m}^-$ still steadily increase with the training epochs, which empirically corroborates the analysis § 5 that REMASKER tends to learn masking-invariant representations.

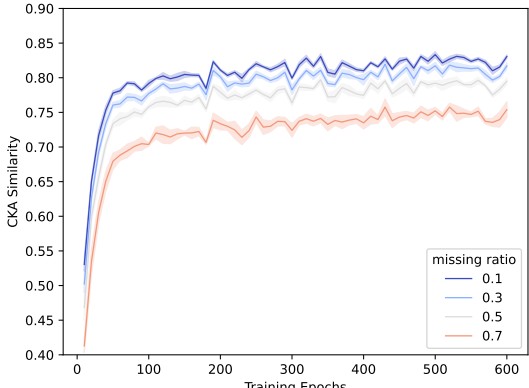

Figure 6: CKA similarity between the representations of inputs under masks $\mathbf{m}^+$ and $\mathbf{m}^-$ (with the number of missing values controlled by the missing ratio). The tested model is trained on `letter` under the MAR setting with 0.5 masking ratio.

| depth | RMSE | WD |
|---|---|---|
| 2 | 0.0783 | 0.0230 |
| 4 | 0.0663 | 0.0172 |
| 6 | 0.0821 | 0.0225 |
| 8 | 0.0834 | 0.0244 |
| 10 | 0.0726 | 0.0196 |

(a) Decoder depth

| width | RMSE | WD |
|---|---|---|
| 16 | 0.0678 | 0.0213 |
| 32 | 0.0663 | 0.0172 |
| 64 | 0.0974 | 0.0322 |
| 128 | 0.1125 | 0.0388 |
| 256 | 0.0877 | 0.0324 |

(b) Embedding width

| depth | RMSE | WD |
|---|---|---|
| 2 | 0.0886 | 0.0334 |
| 4 | 0.0738 | 0.0203 |
| 6 | 0.0663 | 0.0172 |
| 8 | 0.0878 | 0.0322 |
| 10 | 0.0776 | 0.0239 |

(c) Encoder depth

Table 10. Ablation study of REMASKER on the `california` dataset. The default setting is as follows: encoder depth = 6, decoder depth = 4, embedding width = 32, masking ratio = 50%, and training epochs = 600.

## B.4 OVERALL PERFORMANCE

Figure 7, 8, 9, 10, and 11, 12 respectively show the imputation performance of REMASKER and 8 baselines on 12 benchmark datasets under the MAR, MCAR, and MNAR scenarios with the missingness ratio varying from 0.1 to 0.7. Observed that REMASKER performs on par with or outperforms almost all the baselines across a wide range of settings. Note that the MIRACLE imputer does not work on the `Compression` dataset and the `Raisin` dataset under some settings, of which the results are not reported. Given that both `Compression` and `Raisin` are relatively small datasets, one possible explanation is that MIRACLE requires a sufficient amount of data to train the model.

## B.5 SENSITIVITY ANALYSIS

Figure 13 and 14 show the sensitivity analysis of REMASKER and other 6 baselines on the `california` dataset under the MAR, MCAR, and MNAR settings. The observed trends are generally similar to that in Figure 3, which further demonstrates the observations we made in § 4 about how different factors may impact REMASKER's imputation performance.

## B.6 ABLATION STUDY

The ablation study of REMASKER on the `california` dataset is shown in Table 10. Observed that the performance of REMASKER reaches its peak with encoder depth = 6, decoder depth = 4, and embedding width = 32.

## B.7 TRAINING REGIME

Figure 15 shows the imputation performance of REMASKER on the `california` dataset when the training length varies from 100 to 1,600 epochs. Figure 16 plots the convergence of reconstruction loss in REMASKER, showing a trend similar to Figure 4(b).

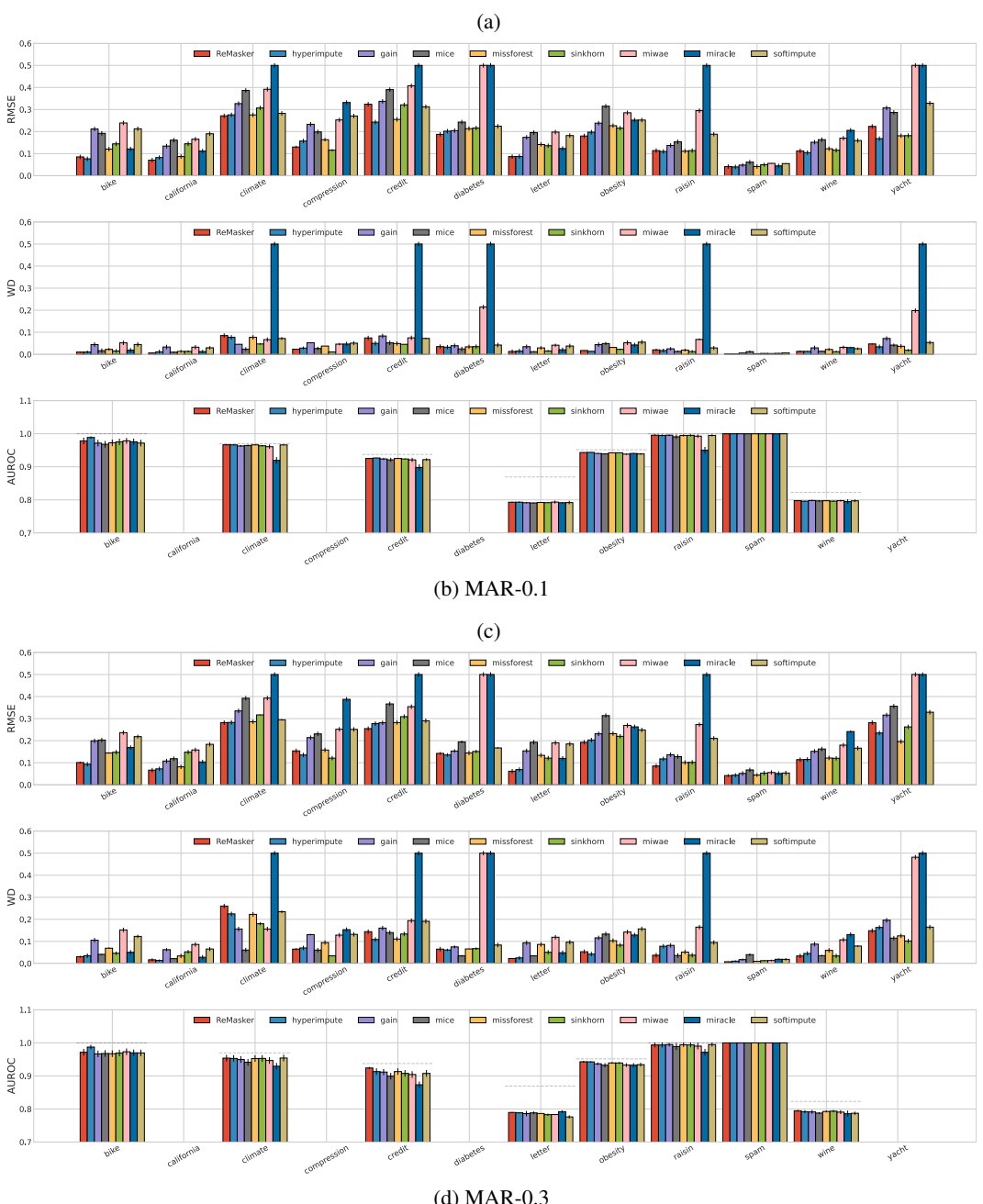

Figure 7: Overall performance of REMASKER and 8 baselines on 12 benchmark datasets under MAR scenario with 0.1 and 0.3 missingness ratio. The results are shown as the mean and standard deviation of RMSE, WD, and AUROC scores (AUROC is only applicable to datasets with classification tasks).

## B.8   OPTIMAL MASKING RATIO VS PAIRWISE MUTUAL INFORMATION

We report below the average pairwise mutual information (PMI) between different features and the performance of ReMasker under varying masking ratios over each dataset. It is observed that in general the PMI value is positively correlated with the optimal masking ratio. This may be explained by that with higher feature correlation (or more information redundancy), a larger masking ratio often leads to more effective representation learning, which corroborates the existing studies on MAE He et al. (2022).

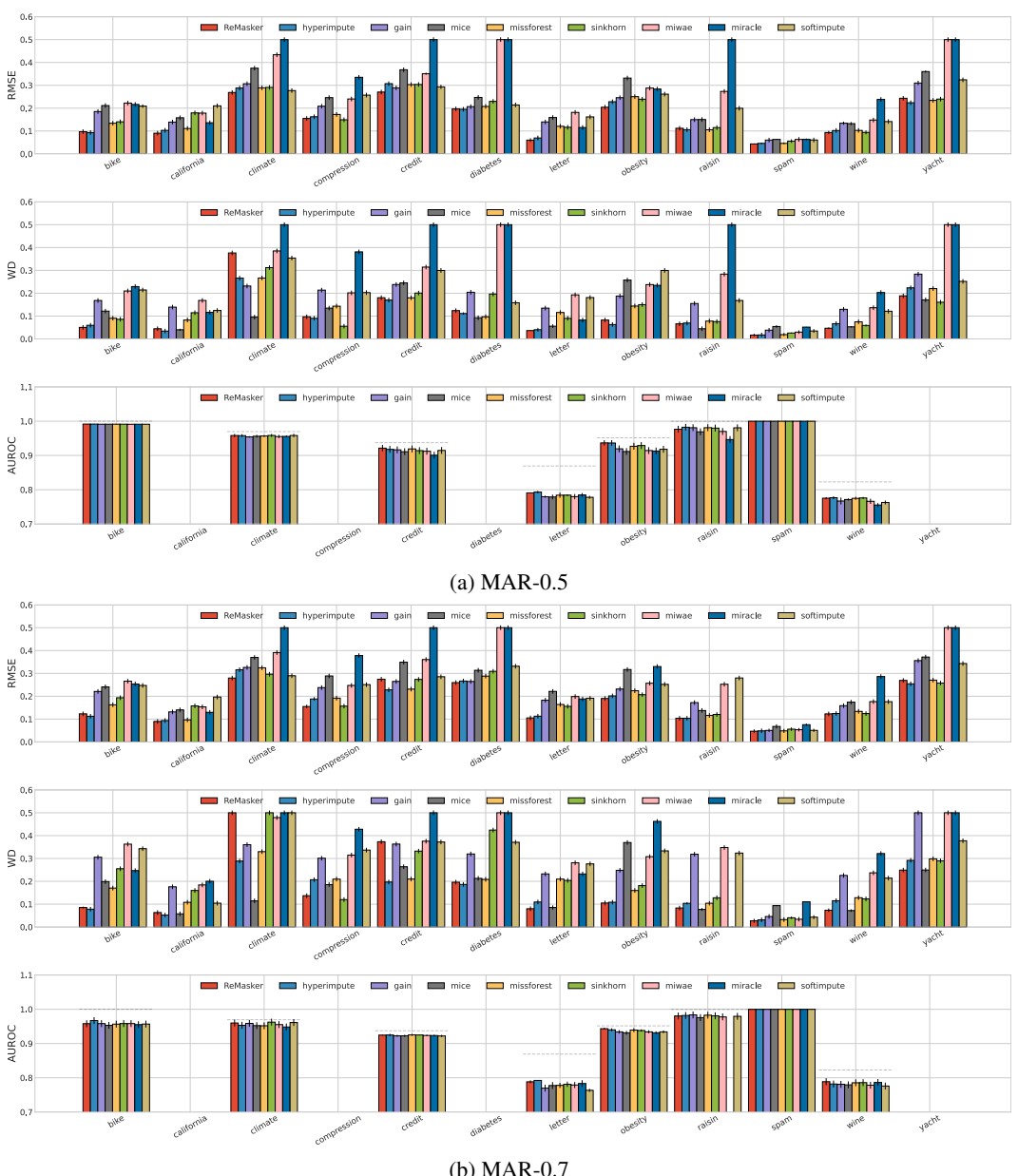

(a) MAR-0.5

(b) MAR-0.7

Figure 8: Overall performance of REMASKER and 8 baselines on 12 benchmark datasets under MAR scenario with 0.5 and 0.7 missingness ratio. The results are shown as the mean and standard deviation of RMSE, WD, and AUROC scores (AUROC is only applicable to datasets with classification tasks).

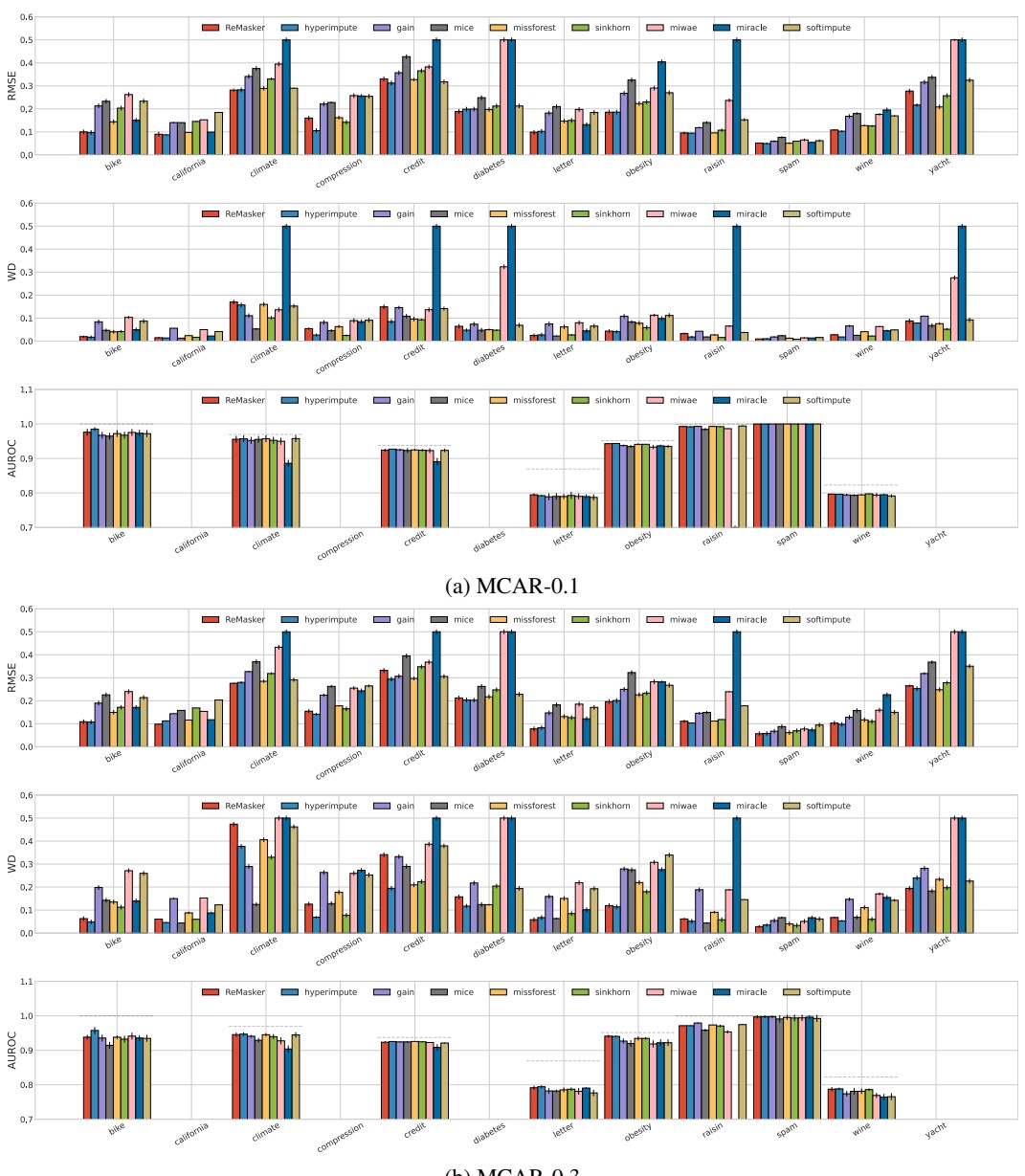

Figure 9: Overall performance of REMASKER and 8 baselines on 12 benchmark datasets under MCAR scenario with 0.1 and 0.3 missingness ratio. The results are shown as the mean and standard deviation of RMSE, WD, and AUROC scores (AUROC is only applicable to datasets with classification tasks).

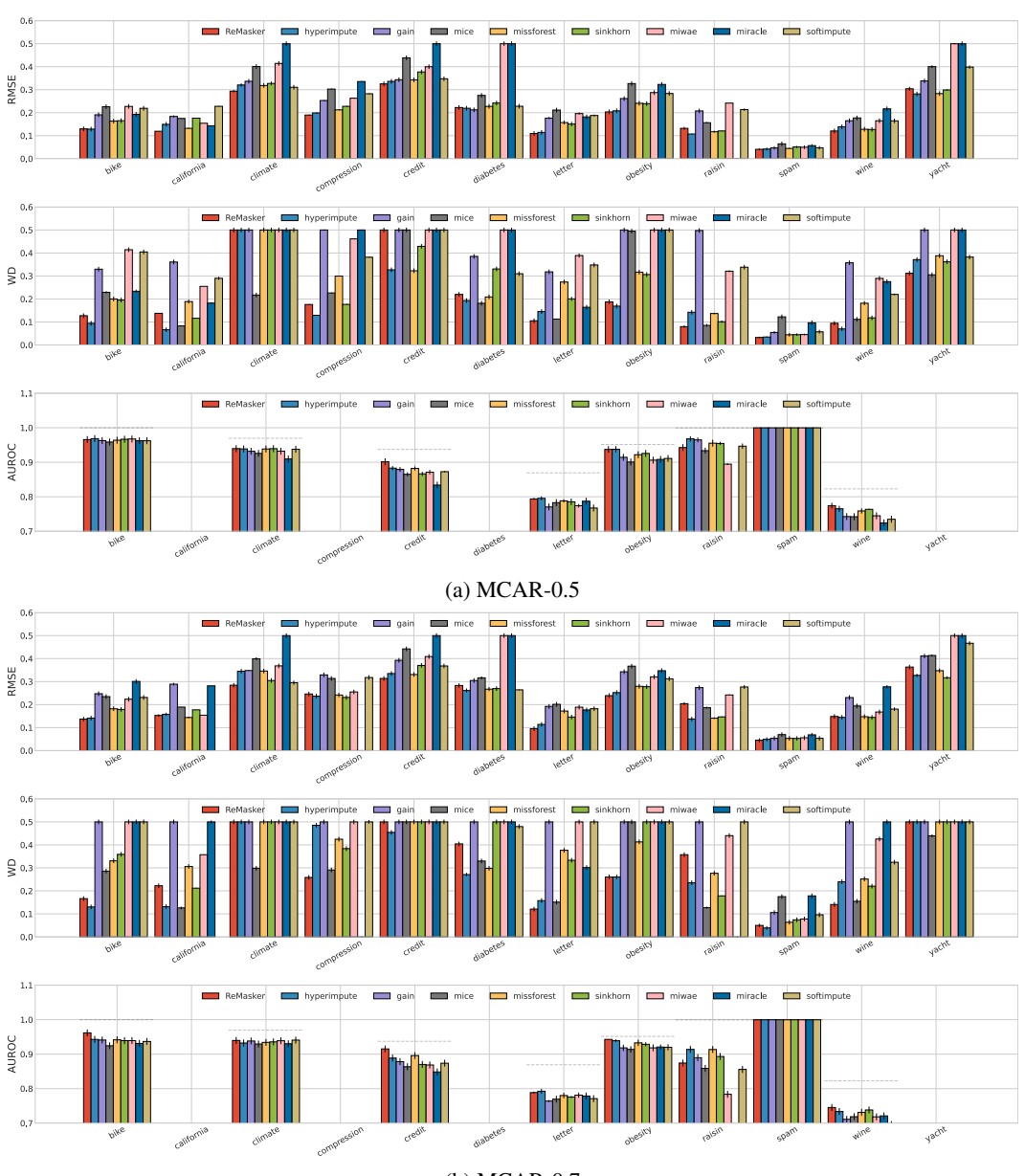

Figure 10: Overall performance of REMASKER and 8 baselines on 12 benchmark datasets under MCAR scenario with 0.5 and 0.7 missingness ratio. The results are shown as the mean and standard deviation of RMSE, WD, and AUROC scores (AUROC is only applicable to datasets with classification tasks).

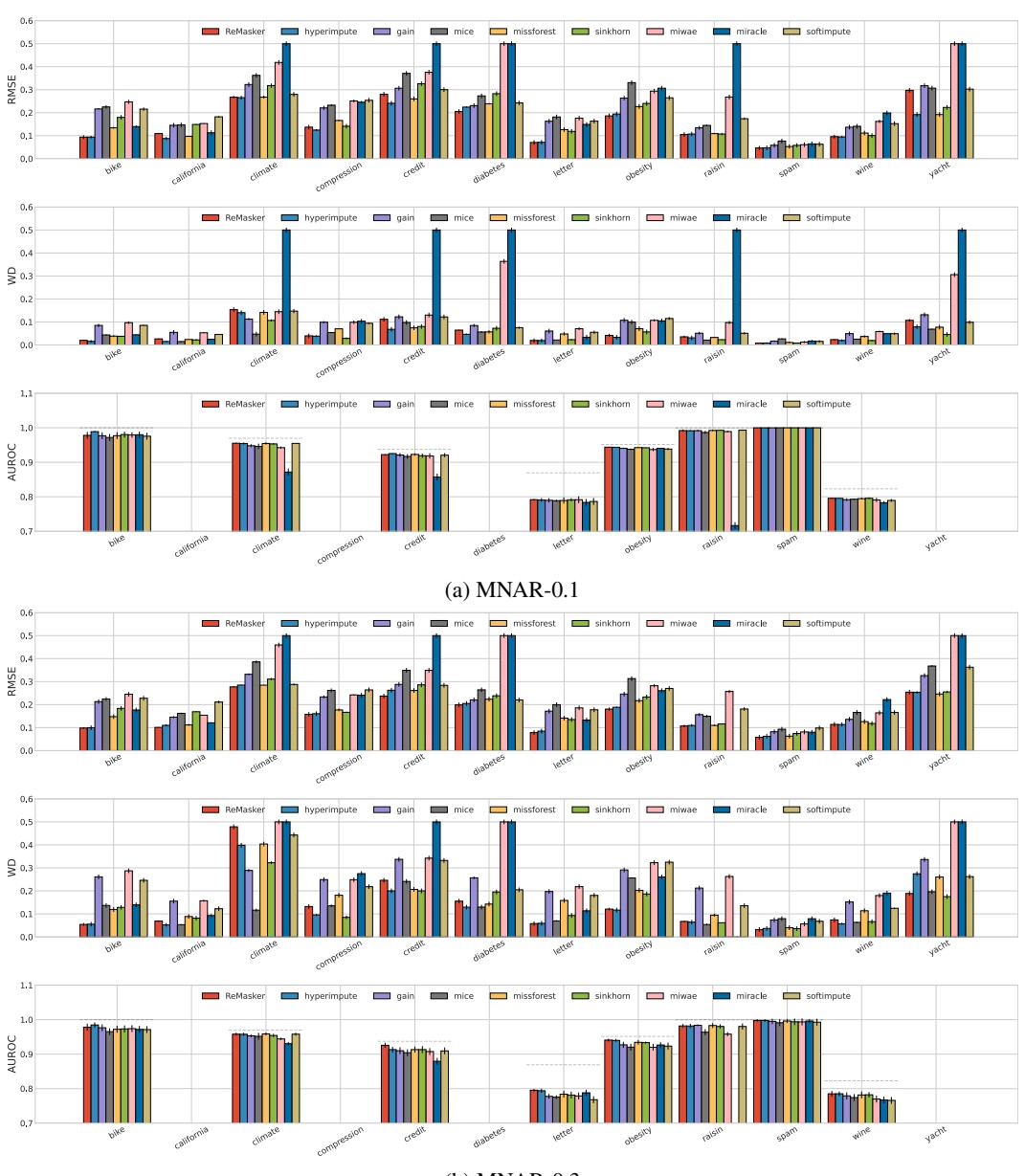

(a) MNAR-0.1

(b) MNAR-0.3

Figure 11: Overall performance of REMASKER and 8 baselines on 12 benchmark datasets under MNAR with 0.1 and 0.3 missingness ratio. The results are shown as the mean and standard deviation of RMSE, WD, and AUROC scores (AUROC is only applicable to datasets with classification tasks).

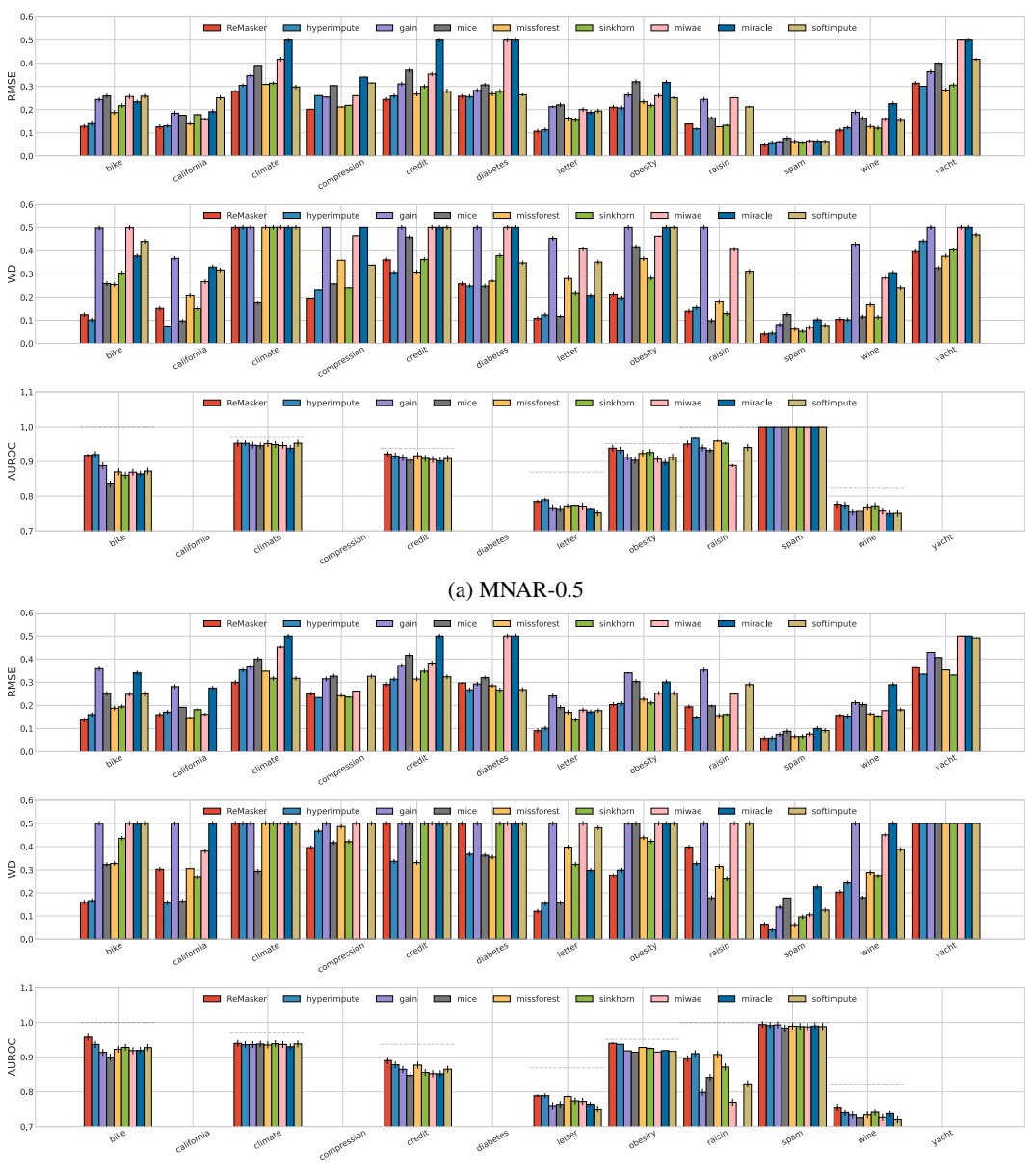

Figure 12: Overall performance of REMASKER and 8 baselines on 12 benchmark datasets under MNAR with 0.5 and 0.7 missingness ratio. The results are shown as the mean and standard deviation of RMSE, WD, and AUROC scores (AUROC is only applicable to datasets with classification tasks).

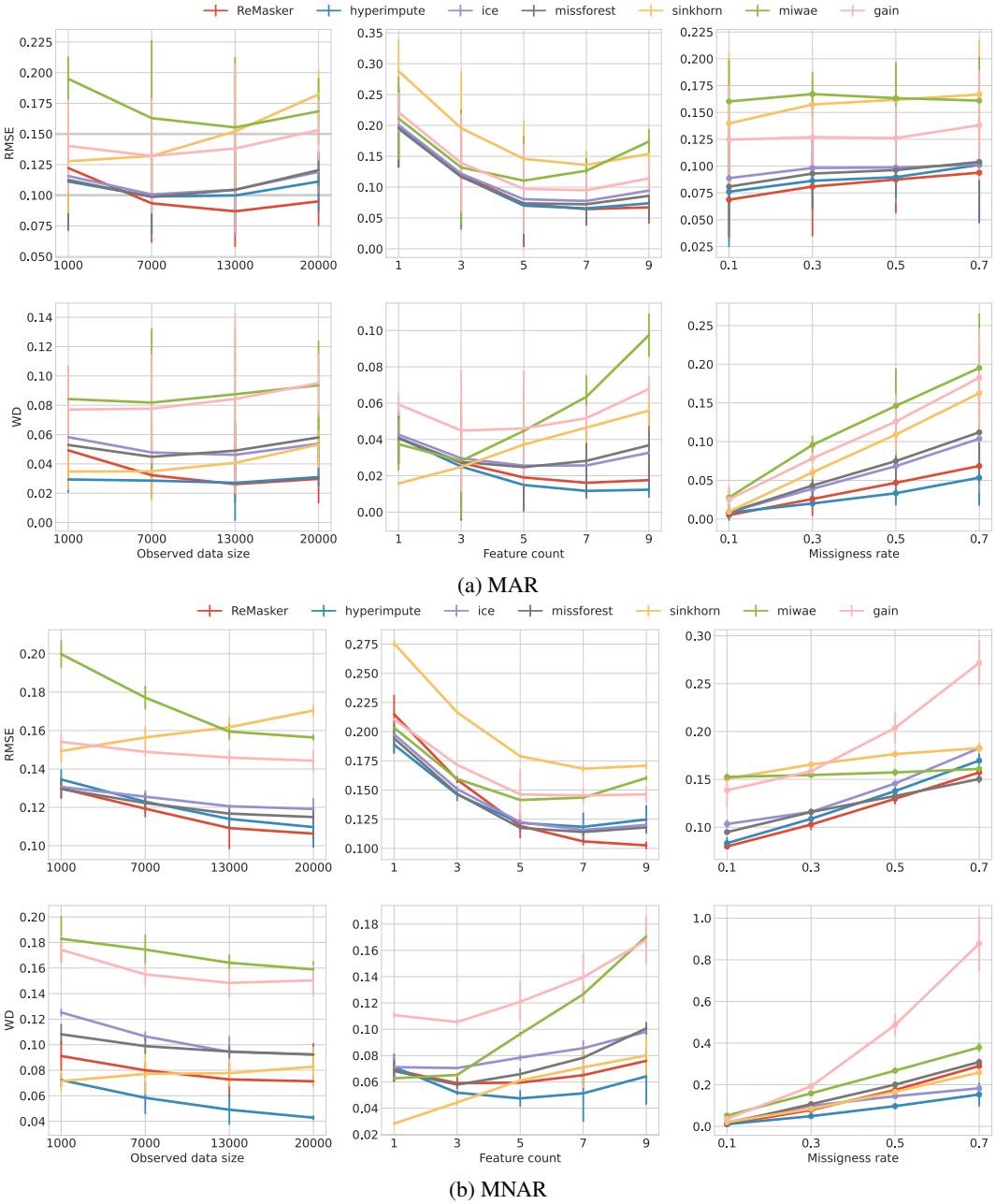

Figure 13: Sensitivity analysis of REMASKER on the `california` dataset under MAR and MNAR scenarios. The results are shown in terms of RMSE and WD, with the scores measured with respect to (a) the dataset size, (b) the number of features, and (c) the missingness ratio. The default setting is as follows: dataset size = 20,000, number of features = 9, and missingness ratio = 0.3.

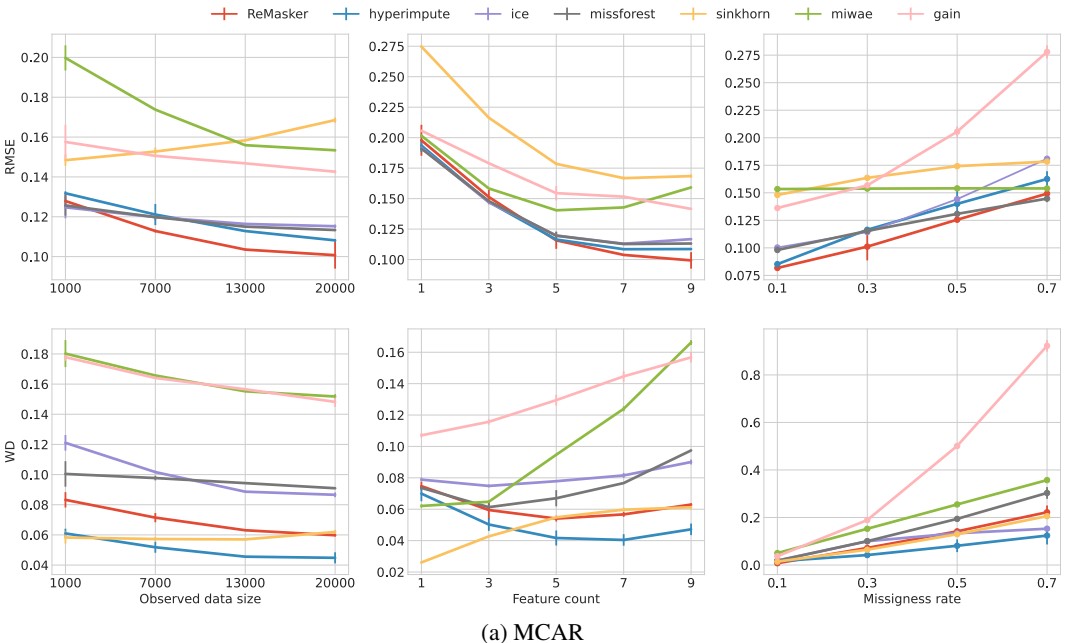

(a) MCAR

Figure 14: Sensitivity analysis of REMASKER on the `california` dataset under the MCAR setting. The results are shown in terms of RMSE and WD, with the scores measured with respect to (a) the dataset size, (b) the number of features, and (c) the missingness ratio. The default setting is as follows: dataset size = 20,000, number of features = 9, and missingness ratio = 0.3.

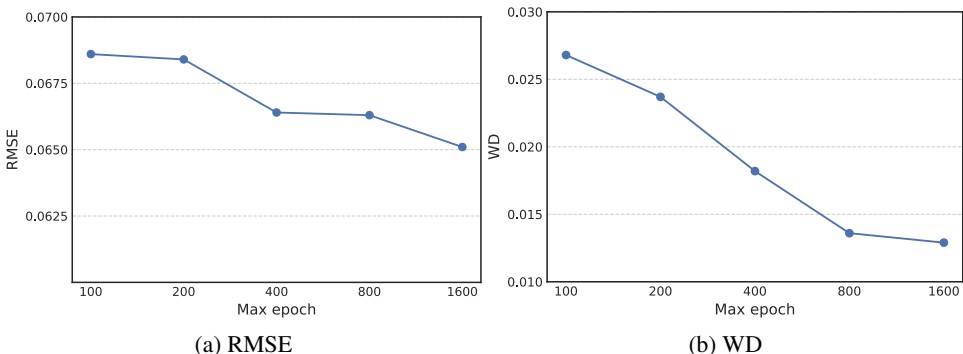

(a) RMSE

(b) WD

Figure 15: REMASKER performance with respect to the number of training epochs on the `california` dataset under MAR with 0.3 missingness ratio.

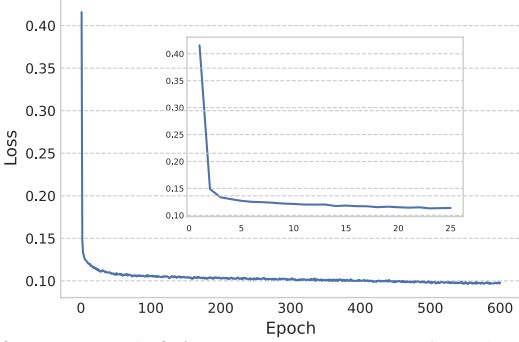

Figure 16: Convergence of REMASKER's fitting on `california` under MAR with 0.3 missingness ratio.

| masking ratio | climate | | | compressive | | diabetes | | obesity | | |
|---|---|---|---|---|---|---|---|---|---|---|
| | RMSE | WD | AUROC | RMSE | WD | RMSE | WD | RMSE | WD | AUROC |
| 0.1 | 0.2825 | 0.2315 | 0.9544 | 0.1739 | 0.0544 | 0.1827 | 0.0691 | 0.2078 | 0.0588 | 0.9411 |
| 0.3 | 0.2832 | 0.2336 | 0.953 | 0.1129 | 0.0439 | 0.1432 | 0.0614 | 0.1943 | 0.0617 | 0.9422 |
| 0.5 | 0.2823 | 0.2481 | 0.9543 | 0.1046 | 0.0306 | 0.1496 | 0.0694 | 0.193 | 0.0743 | 0.9417 |
| 0.7 | 0.2837 | 0.2544 | 0.9533 | 0.1635 | 0.0825 | 0.1542 | 0.0784 | 0.2129 | 0.1093 | 0.9408 |
| PMI | 0.023 | | | 0.7829 | | 0.1106 | | 0.147 | | |

| masking ratio | credit | | | wine | | | raisin | | | spam | | |
|---|---|---|---|---|---|---|---|---|---|---|---|---|
| | RMSE | WD | AUROC | RMSE | WD | AUROC | RMSE | WD | AUROC | RMSE | WD | AUROC |
| 0.1 | 0.2298 | 0.1075 | 0.9255 | 0.1171 | 0.0255 | 0.7954 | 0.1497 | 0.0727 | 0.9931 | 0.0451 | 0.0097 | 0.9998 |
| 0.3 | 0.2277 | 0.0993 | 0.9279 | 0.1122 | 0.0303 | 0.7949 | 0.0917 | 0.0431 | 0.9929 | 0.0434 | 0.0084 | 0.9998 |
| 0.5 | 0.245 | 0.1243 | 0.9247 | 0.1151 | 0.0395 | 0.7955 | 0.0904 | 0.0347 | 0.9943 | 0.0435 | 0.0099 | 0.9998 |
| 0.7 | 0.3175 | 0.2176 | 0.914 | 0.1401 | 0.0757 | 0.7934 | 0.1172 | 0.0727 | 0.9946 | 0.0507 | 0.0197 | 0.9998 |
| PMI | 0.069 | | | 0.1612 | | | 0.5246 | | | 0.0771 | | |

| masking ratio | bike | | | yacht | | letter | | | california | |
|---|---|---|---|---|---|---|---|---|---|---|
| | RMSE | WD | AUROC | RMSE | WD | RMSE | WD | AUROC | RMSE | WD |
| 0.1 | 0.1068 | 0.0244 | 0.97 | 0.2882 | 0.1339 | 0.0668 | 0.0215 | 0.0789 | 0.0888 | 0.023 |
| 0.3 | 0.096 | 0.0226 | 0.9736 | 0.2185 | 0.0812 | 0.0562 | 0.0207 | 0.7897 | 0.0654 | 0.0151 |
| 0.5 | 0.1015 | 0.0277 | 0.9707 | 0.2174 | 0.0996 | 0.0554 | 0.0212 | 0.7935 | 0.0663 | 0.0172 |
| 0.7 | 0.1306 | 0.0561 | 0.9687 | 0.3188 | 0.1948 | 0.0906 | 0.0366 | 0.7878 | 0.132 | 0.065 |
| PMI | 0.1614 | | | 0.8229 | | 0.1421 | | | 0.1576 | |

Table 11. REMASKER performance with respect to masking ratio. The results are evaluated under MAR with 0.3 missingness ratio.

