# OpenReview forum: "ReMasker: Imputing Tabular Data with Masked Autoencoding"
_ICLR.cc/2024/Conference — ICLR 2024 poster_

### Official Review · Reviewer_YQFA · 2023-10-25

**Soundness:** 2 fair
**Presentation:** 3 good
**Contribution:** 2 fair
**Rating:** 5
**Confidence:** 3

**Summary:**

The paper presents an imputation method, ReMasker, that leverages a remasking autoencoder with a Transformer-based architecture. A significant amount of experimentation is included to validate the proposed procedure.

**Strengths:**

The proposed method provides a simpler heuristic than competing SOTA methods such as HyperImpute, while displaying similar or slightly improved accuracy.

The experiments are relatively thorough and show strong results for the MCAR setting in particular.

**Weaknesses:**

The methodology is not novel, as masking is a common practice for imputation algorithms and masked autoencoders with Transformers are used for other tasks. However, the authors do not claim novelty and I agree this is likely the first work to consider a masked autoencoder with Transformers for imputation specifically.

The claims that ReMasker generalizes to MNAR settings is not well justified:

* It’s not clear how we could properly learn a missingness-invariant representation in the MNAR setting. Under MNAR, the missing data values fundamentally depend on the missingness mechanism, so recovering a missingness-invariant representation would require external information.

* The experimental condition for MNAR (masking via a Bernoulli random variable with fixed mean) is not representative of many MNAR settings. For example, cases where values across possibly many variables are masked due to censoring of one or more variables (e.g. $X_2$, $X_5$, $X_6$ are missing when $X_5 < 0$). The missing values often follow a completely different distribution than the observed values, which cannot be recovered using the observed data alone.

It is not clear that the results are practically different from HyperImpute, and perhaps other methods. The included bar plots are hard to read, so any assessment is difficult. The relative simplicity of ReMasker compared to HyperImpute is mentioned as one advantage, although the much simpler ICE appears competitive in almost all scenarios.

While the theoretical justification is appropriately included in the Discussion section, the justification is a conjecture with minimal validation. In practice, the missingness mechanism is different for $m^+$ and $m^-$, so it is not clear that representation will be invariant in general. Further experimentation is needed.

**Questions:**

Is there a better way to represent the results visually? It is very difficult to read the plots (e.g. Figure 2), and in particular to the confidence intervals.

While it is clearly stated that imputation for downstream tasks is not part of the scope of the paper, can any more discussion be provided? Considering imputation is almost always followed by downstream tasks, I believe more justification for this decision is needed.

---

> ### Author Response · Authors · 2023-11-15
>
> We thank the reviewer for the valuable feedback on improving this paper! Please find below our response to the reviewer’s questions.
>
> > The methodology is not novel, as masking is a common practice for imputation algorithms and masked autoencoders with Transformers are used for other tasks. However, the authors do not claim novelty and I agree this is likely the first work to consider a masked autoencoder with Transformers for imputation specifically.
>
> While both Transformer and masked autoencoding (MAE) have been used in prior work, this work adapts the MAE approach (with Transformer as the backbone) in a nuanced yet novel way: besides the missing values in the given dataset, we randomly select and “re-mask” another set of values, optimize the autoencoder with the objective of reconstructing this re-masked set, and then apply the trained autoencoder to predict the missing values. This resulting re-masking approach serves as a strong baseline for developing and evaluating future imputation methods.
>
> > The claims that ReMasker generalizes to MNAR settings is not well justified:
> > * It’s not clear how we could properly learn a missingness-invariant representation in the MNAR setting. Under MNAR, the missing data values fundamentally depend on the missingness mechanism, so recovering a missingness-invariant representation would require external information.
> > * The experimental condition for MNAR (masking via a Bernoulli random variable with fixed mean) is not representative of many MNAR settings. For example, cases where values across possibly many variables are masked due to censoring of one or more variables (e.g. X_2, X_5, X_6 are missing when X_5<0). The missing values often follow a completely different distribution than the observed values, which cannot be recovered using the observed data alone.
>
> We thank the reviewer for the insightful comments. In general, it is impossible to identify the missingness distribution in MNAR without domain-specific assumptions or constraints (Ma & Zhang, 2021). That is why most prior work (e.g., Yong et al., 2019; Yoon and Sull, 2020; Jarrett et al., 2022) focuses on MCAR and MAR. For completeness, following the setting of Jarrett et al., (2022), we also evaluate ReMasker under one specific MNAR setting and show that it also partially generalizes to such settings. We acknowledge that there are MNAR settings (such as the ones suggested by the reviewer) where access to external knowledge is essential. We have revised the statement (**Section 4**) to more accurately reflect our work’s focus on MAR and MCAR and its limitations with respect to MNAR.
>
> Further, we have added a detailed discussion about how ReMasker’s performance is influenced by the missingness mechanism (**Section 5 Q2**). Intuitively, ReMasker encourages learning representations invariant to re-masked values and then leverages such representations to impute missing values. Thus, while ReMasker takes effect as long as the distributions of re-masked and missing values are correlated, its performance may vary with the disparity between the two distributions. In MACR, the distributions of re-masked and missing values are highly similar; in MAR, the distribution of re-masked values is biased towards the observable values; in MNAR, there is even more disparity between the two distributions. Thus, ReMasker’s performance tends to vary as MCAR > MAR > MNAR.

---

> > ### Author Response · Authors · 2023-11-15
> >
> > > It is not clear that the results are practically different from HyperImpute, and perhaps other methods. The included bar plots are hard to read, so any assessment is difficult. The relative simplicity of ReMasker compared to HyperImpute is mentioned as one advantage, although the much simpler ICE appears competitive in almost all scenarios.
> >
> > To better visualize the improvement of ReMasker over other baselines, we have summarized the performance of ReMasker and baselines (measured and ranked in terms of RMSE) on 12 benchmark datasets under MAR with 0.3 missingness ratio in Table 10 (**Appendix B3**). The key observation is that ReMasker consistently outperforms other baselines across most datasets (ranked top 1 or 2). Compared with HyperImpute, ReMasker is a much simpler and more modular imputer; therefore, it may serve as a strong baseline to develop or evaluate future imputation methods or it can be integrated into an ensemble imputer such as HyperImpute. In addition, we further compare the running time of ReMasker (under the default setting of Table 7) and HyperImpute on datasets with over 1,000 records, with results summarized as follows (**Appendix B.2**). Observe that compared with HyperImpute, ReMasker is not only more scalable but also less sensitive to data size.
> >
> > | | compression |  wine |  spam | credit | bike | obesity | california |
> > | :----: | :----: | :----: | :----: | :----: | :----: | :----: | :----: |
> > | **HyperImpute** | 154.9s | 297.7s | 185.8s | 39.9s | 259.9s | 196.1s | 195.8s |
> > | **ReMasker** | 72.9s | 73.8s | 74.1s | 74.0s | 66.9s | 66.8s |66.9s |
> >
> > > While the theoretical justification is appropriately included in the Discussion section, the justification is a conjecture with minimal validation. In practice, the missingness mechanism is different for m+ and m-, so it is not clear that representation will be invariant in general. Further experimentation is needed.
> >
> > To further validate the masking-invariant properties of ReMasker, we measure the CKA similarity (Kornblith et al., 2019) between the latent representations (i.e., the output of ReMasker’s encoder) of inputs under the masks of $m^+$ and $m^-$, with results shown in Figure 6 (**Appendix B4**). Observe that the CKA measures between the inputs under the two masks steadily increase with the training epochs, which empirically corroborates the analysis in Section 5 Q1 that ReMasker tends to learn representations that are invariant to various missingness.
> >
> > > While it is clearly stated that imputation for downstream tasks is not part of the scope of the paper, can any more discussion be provided? Considering imputation is almost always followed by downstream tasks, I believe more justification for this decision is needed.
> >
> > We thank the reviewer for the question. Following the line of work on imputing tabular data (e.g., Mattei & Frellsen, 2018; Yong et al., 2019; Nazabal et al., 2020; Yoon and Sull, 2020; Jarrett et al., 2022), we do not consider imputing missing values as a means to obtain data for known downstream tasks. Under those settings, the imputation is fundamentally entangled with the downstream task, typically requiring joint training to optimize the downstream task and the imputation task simultaneously. Due to the potential conflict between the objectives of the two tasks (i.e., data fidelity versus downstream-task performance), it is not straightforward to evaluate the performance of imputation methods directly. We thus focus on the imputation task with data fidelity as the main metric.
> >
> > Again, we thank the reviewer for the valuable feedback. Please let us know if there are any other questions or suggestions.
> >
> > Best,
> >
> > Authors

---

> > > ### Comment · Reviewer_YQFA · 2023-11-20
> > >
> > > > To better visualize the improvement of ReMasker over other baselines, we have summarized the performance of ReMasker and baselines (measured and ranked in terms of RMSE) on 12 benchmark datasets under MAR with 0.3 missingness ratio in Table 10 (Appendix B3).
> > >
> > > Thank you, this addition is greatly appreciated.
> > >
> > > > The key observation is that ReMasker consistently outperforms other baselines across most datasets (ranked top 1 or 2).
> > >
> > > I acknowledge that ReMasker is consistently the best performing algorithm in your experiments. However, based on the provided standard deviation estimates, the improvement is not significant. It is not clear whether the improvement is practically significant for downstream tasks that leverage the imputed data.
> > >
> > > > Compared with HyperImpute, ReMasker is a much simpler and more modular imputer; therefore, it may serve as a strong baseline to develop or evaluate future imputation methods or it can be integrated into an ensemble imputer such as HyperImpute.
> > >
> > > I agree with this observation, and for this reason in particular, I believe ReMasker is a potentially valuable contribution.
> > >
> > > > To further validate the masking-invariant properties of ReMasker, we measure the CKA similarity (Kornblith et al., 2019) between the latent representations (i.e., the output of ReMasker’s encoder) of inputs under the masks of $m^+$ and $m^-$, with results shown in Figure 6 (Appendix B4). Observe that the CKA measures between the inputs under the two masks steadily increase with the training epochs, which empirically corroborates the analysis in Section 5 Q1 that ReMasker tends to learn representations that are invariant to various missingness.
> > >
> > > This addition is greatly appreciated, and I agree it does add evidence about the paper's claims. Can you clarify how the missingness mechanisms for $m^+$ and $m^-$ differ in this experiment?
> > >
> > > > We thank the reviewer for the question. Following the line of work on imputing tabular data (e.g., Mattei & Frellsen, 2018; Yong et al., 2019; Nazabal et al., 2020; Yoon and Sull, 2020; Jarrett et al., 2022), we do not consider imputing missing values as a means to obtain data for known downstream tasks. Under those settings, the imputation is fundamentally entangled with the downstream task, typically requiring joint training to optimize the downstream task and the imputation task simultaneously. Due to the potential conflict between the objectives of the two tasks (i.e., data fidelity versus downstream-task performance), it is not straightforward to evaluate the performance of imputation methods directly. We thus focus on the imputation task with data fidelity as the main metric.
> > >
> > > This is reasonable. I apologize that my original question was relatively vague, I would like to provide some additional context. You compare your results with algorithms such as MICE, a multiple imputation framework that was not originally intended for SOTA imputation accuracy necessarily, but rather the ability to measure the added uncertainty from missing data in downstream *inferential* tasks. Based on your justification, I believe the focus of the paper is reasonable, but although I encourage the authors to consider the value of ReMasker with a focus on improved uncertainty estimation in downstream tasks, possibly as part of future research.

---

> > > ### Comment · Reviewer_YQFA · 2023-11-20
> > > **Thank you**
> > >
> > > I would like to thank the authors for taking the time to reply to the various comments from myself and my fellow reviewers. I believe the revised submission is an improvement over the previous version. In particular, the improved exposition surrounding MNAR and the addition of appendices B1-B4.
> > >
> > > I still have significant concerns about novelty and improvement over other baselines, but I agree with the authors that ReMasker serves as a simple baseline for future work or as part of future ensemble methods.
> > >
> > > For those reasons, I will update my score to a 5.

---

> > > > ### Author Response · Authors · 2023-11-21
> > > >
> > > > We thank the reviewer for the valuable feedback and follow-up! We are so delighted to learn that our response and additional experiments have addressed most of your queries. Please see our response to your additional questions below.
> > > >
> > > > ---
> > > >
> > > > >> While both Transformer and masked autoencoding (MAE) have been used in prior work, this work adapts the MAE approach (with Transformer as the backbone) in a nuanced yet novel way: besides the missing values in the given dataset, we randomly select and “re-mask” another set of values, optimize the autoencoder with the objective of reconstructing this re-masked set, and then apply the trained autoencoder to predict the missing values. This resulting re-masking approach serves as a strong baseline for developing and evaluating future imputation methods.
> > > >
> > > > > Unfortunately I still have concerns about novelty. For instance, masking observed data has been previously studied for missing data problems, for example Gondara and Wang, 2018, where they refer to masking as stochastic corruption.
> > > >
> > > > We thank the reviewer for the insightful comments. In the revised manuscript, we have included a brief discussion of this missing reference in **Section 5 Q1**. While the idea of masking observable data is not completely new, to our best knowledge, this work is the first to adapt it in the context of masked autoencoding (that is why the name "re-masking" is dubbed). This leads to a somewhat more elegant imputation method. For example, as the denoising autoencoder requires complete data to operate, Gondara and Wang (2018) need to set the respective column average as placeholders for missing values and zeros for masked values at initialization. In comparison, ReMasker does not require initializing missing or re-masked values in the input, while representing them with the same trainable tokens in the latent space encourages learning the distribution of re-masked values close to the distribution of missing values
> > > >
> > > > ---
> > > >
> > > > >> To further validate the masking-invariant properties of ReMasker, we measure the CKA similarity (Kornblith et al., 2019) between the latent representations (i.e., the output of ReMasker’s encoder) of inputs under the masks of $m^+$ and $m^-$, with results shown in Figure 6 (Appendix B4). Observe that the CKA measures between the inputs under the two masks steadily increase with the training epochs, which empirically corroborates the analysis in Section 5 Q1 that ReMasker tends to learn representations that are invariant to various missingness.
> > > >
> > > > > This addition is greatly appreciated, and I agree it does add evidence about the paper's claims. Can you clarify how the missingness mechanisms for and differ in this experiment?
> > > >
> > > > Given $m$ as the mask of missing values and $m'$ as the mask of re-masked values, $m^+ = m \odot m'$ ($\odot$ denotes element-wise multiplication) masks out both missing and re-masked values, while $m^- = m \odot (1-m')$ masks out missing values and values that are not re-masked.
> > > >
> > > > ---
> > > >
> > > > >> We thank the reviewer for the question. Following the line of work on imputing tabular data (e.g., Mattei & Frellsen, 2018; Yong et al., 2019; Nazabal et al., 2020; Yoon and Sull, 2020; Jarrett et al., 2022), we do not consider imputing missing values as a means to obtain data for known downstream tasks. Under those settings, the imputation is fundamentally entangled with the downstream task, typically requiring joint training to optimize the downstream task and the imputation task simultaneously. Due to the potential conflict between the objectives of the two tasks (i.e., data fidelity versus downstream-task performance), it is not straightforward to evaluate the performance of imputation methods directly. We thus focus on the imputation task with data fidelity as the main metric.
> > > >
> > > > > This is reasonable. I apologize that my original question was relatively vague, I would like to provide some additional context. You compare your results with algorithms such as MICE, a multiple imputation framework that was not originally intended for SOTA imputation accuracy necessarily, but rather the ability to measure the added uncertainty from missing data in downstream inferential tasks. Based on your justification, I believe the focus of the paper is reasonable, but although I encourage the authors to consider the value of ReMasker with a focus on improved uncertainty estimation in downstream tasks, possibly as part of future research.
> > > >
> > > > We thank the reviewer's comments. We have added improving uncertainty estimation in downstream tasks as one of future research focuses in **Section 5 Q4**.
> > > >
> > > > ---
> > > >
> > > > Again, we thank the reviewer for the valuable feedback. Please let us know if there are any other questions or suggestions.
> > > >
> > > > Best,
> > > >
> > > > Authors

---

> > ### Comment · Reviewer_YQFA · 2023-11-20
> >
> > > While both Transformer and masked autoencoding (MAE) have been used in prior work, this work adapts the MAE approach (with Transformer as the backbone) in a nuanced yet novel way: besides the missing values in the given dataset, we randomly select and “re-mask” another set of values, optimize the autoencoder with the objective of reconstructing this re-masked set, and then apply the trained autoencoder to predict the missing values. This resulting re-masking approach serves as a strong baseline for developing and evaluating future imputation methods.
> >
> > Unfortunately I still have concerns about novelty. For instance, masking observed data has been previously studied for missing data problems, for example Gondara and Wang, 2018, where they refer to masking as stochastic corruption.
> >
> > > We have revised the statement (Section 4) to more accurately reflect our work’s focus on MAR and MCAR and its limitations with respect to MNAR.
> >
> > I believe the revised exposition for the MNAR setting in particular is much more reasonable and a significant improvement.

---

### Official Review · Reviewer_Fz3c · 2023-10-30

**Soundness:** 3 good
**Presentation:** 3 good
**Contribution:** 3 good
**Rating:** 6
**Confidence:** 3

**Summary:**

The authors propose ReMasker, a method for imputing missing values in tabular data based on masked auto encoders (MAE). Specifically, given a dataset, they mask some of the features and train a model to impute these features. The approach does not require all features to be present in training, so a dataset with missing values can be used for training. The authors provide extensive empirical study and a theoretical justification.

**Strengths:**

I think the authors propose an elegant solution to a problem many practitioners encounter. The paper is well written and the empirical results are convincing.
In general, for an approach such as the one presented, it is difficult to provide a theoretical analysis. I appreciate the theoretical justification in Section 5, but I would have liked it to have a more prominent position in the paper.

**Weaknesses:**

The authors fail to mention and discuss multiple previous works. Specifically, transformers have been previously been applied to tabular data (Somepalli et al. 2021, Arık et al. 2020,  Huang et al 2020). Specific to Huang et al, the authors cite this paper for supporting their model choice, but fail to mention it in the related works. The MAE work by Majmundar et al. (2022) is also not discussed.


Somepalli et al. 2021: SAINT: Improved Neural Networks for Tabular Data via Row Attention and Contrastive Pre-Training
Arık et al. 2020: TabNet: Attentive Interpretable Tabular Learning
Huang et al 2020: TabTransformer: Tabular Data Modeling Using Contextual Embeddings
Majmundar et al. 2022: MET: Masked Encoding for Tabular Data

**Questions:**

I have one minor question:
- On page 1, you state that discriminative imputers are hindered by the requirement of specifying a functional form. It seems to me that your architecture is also a choice of "functional form". Am I wrong?

---

> ### Author Response · Authors · 2023-11-15
>
> We thank the reviewer for the valuable feedback on improving this paper! Please find below our response to the reviewer’s questions.
>
> > The authors fail to mention and discuss multiple previous works. Specifically, transformers have been previously been applied to tabular data (Somepalli et al. 2021, Arık et al. 2020, Huang et al 2020). Specific to Huang et al, the authors cite this paper for supporting their model choice, but fail to mention it in the related works. The MAE work by Majmundar et al. (2022) is also not discussed.
>
> We thank the reviewer for pointing out the missing references. We have included a discussion of these works (**Section 2**). While SAINT, TabNet, and TabTransformer also use Transformer as the backbone to model tabular data, the main idea of ReMasker is to adapt the masked autoencoding (MAE) approach to imputing missing values of tabular data: besides the missing values in the given dataset, randomly select and “re-mask” another set of values, optimize the autoencoder with the objective of reconstructing this re-masked set, and then apply the trained autoencoder to predict the missing values. Meanwhile, although MET also employs the MAE approach, its primary focus is on the representation learning of tabular data, operating under the assumption of data completeness and applying MAE in a relatively direct manner. In contrast, ReMasker, with its emphasis on imputing missing values, inherently assumes incomplete data and adapts the MAE technique in a nuanced yet novel way.
>
> > On page 1, you state that discriminative imputers are hindered by the requirement of specifying a functional form. It seems to me that your architecture is also a choice of "functional form". Am I wrong?
>
> We thank the reviewer for the question. We would like to clarify that the functional form refers to the conditional distributions of missing values. Specifically, the discriminative methods often impute missing values by modeling their conditional distributions on the basis of observable values, therefore these methods are hindered by the requirement of specifying the proper functional forms about the conditional distributions.
>
> Again, we thank the reviewer for the valuable feedback. Please let us know if there are any other questions or suggestions.
>
> Best,
>
> Authors

---

> > ### Comment · Reviewer_Fz3c · 2023-11-20
> > **Thank you for your Updates**
> >
> > I thank the authors for updating their paper with a more detailed discussion of existing works. Since I believe that the proposed method is valuable to the community, and after reading your answer to all reviewers, I would like to increase my score to 6.

---

> > > ### Author Response · Authors · 2023-11-21
> > >
> > > We thank the reviewer for the valuable feedback and follow-up! We are so delighted to learn that our response has addressed your queries. We are committed to incorporating all the comments from you and the other reviewers in our revised manuscript.
> > >
> > > Best,
> > >
> > > Authors

---

### Official Review · Reviewer_w2Hd · 2023-10-31

**Soundness:** 3 good
**Presentation:** 3 good
**Contribution:** 3 good
**Rating:** 8
**Confidence:** 4

**Summary:**

The paper Remasker: Imputing Tabular Data with Masked Autoencoding presents a novel algorithm for missing data imputation in tabular datasets using the masked autoencoding framework. The algorithm operates in two stages: a first stage where extra variables are masked and the autoencoder is optimized to reconstruct the re-masked variables, and a second stage where the trained model is used to predict the missing values. Authors evaluate the method on a wide range of state-of-the-art methods in several tabular datasets, showing competitive performance.

**Strengths:**

The paper is well-written and grammatically correct with no evident typos. Every section is very well outlined with clear paragraphs describing the content of the manuscript.

The proposed method, namely the Remasker method, is easy to understand for experts and non-experts in the field. The idea is intuitive yet effective, showing promising results on how we can leverage the promises of transformer-based architecture for missing data imputation.

The evaluation of the method is quite thorough, where standard tabular datasets are tested under different state-of-the-art methods. This reveals that the authors are aware of the most recent advances in missing data imputation. Besides, different metrics and ablation studies are performed to motivate the performance of the method further. Another precious point is that the authors do comment on the limitations of their proposed method, which is something that is sometimes overlooked.

**Weaknesses:**

I have a few concerns and questions that I'd like the authors to address:

1. While I understand the primary motivation behind using a transformer-based architecture in this work, I would appreciate a more in-depth discussion of why the authors chose to use masked autoencoders over other approaches, like deep generative models. It's crucial to outline the advantages and disadvantages of this choice. Additionally, it would be beneficial to address questions like the computational cost of training the Remasker method and its ability to generate new data, which is a common concern in the missing data imputation field.
2. The argument presented in Section 4.1 about why Remasker generalizes different missing data assumptions could be strengthened. While transformer-based architectures have proven their superior performance in various tasks, a more comprehensive analysis of why transformers excel in missing data imputation would enhance the paper's quality and its contribution to both the missing data imputation and transformer-based research fields.
3. The manuscript appears to be densely packed in terms of space. Figures and tables are positioned very closely to the text, which can make the paper seem cluttered and condensed. Consider the possibility of making some tables smaller or relocating them to the Appendix (e.g., Table 1), and possibly reducing the size of certain figures (e.g., Figure 2 with fewer datasets). This issue can be addressed in the camera-ready version if the paper is accepted.

**Questions:**

Some questions I have are the following:

1. I'd like to understand the rationale behind using a deep encoder and a shallow decoder when describing the decoder. While the Ablation Study in Section 4.3 provides some insight, could you provide a more abstract explanation for this choice? Is there a specific reason why you believe the first fitting stage of the Remasker benefits from a more complex encoder?
2. Some figures display results for different baselines (e.g., Figure 3 RMSE), and while they may appear distinct, the values on the y-axis are actually quite close. Could you elaborate on how these slight differences influence the final results? For instance, in Figure 3, Remasker outperforms other methods by only a narrow margin in terms of RMSE and AUROC. How does this marginal improvement make Remasker a more beneficial choice compared to other methods that utilize deep generative models to learn the data distribution?

---

> ### Author Response · Authors · 2023-11-15
>
> We thank the reviewer for the valuable feedback on improving this paper! Please find below our response to the reviewer’s questions.
>
> > While I understand the primary motivation behind using a transformer-based architecture in this work, I would appreciate a more in-depth discussion of why the authors chose to use masked autoencoders over other approaches, like deep generative models. It's crucial to outline the advantages and disadvantages of this choice. Additionally, it would be beneficial to address questions like the computational cost of training the Remasker method and its ability to generate new data, which is a common concern in the missing data imputation field.
>
> Our use of masked autoencoding (MAE) is motivated by its effectiveness of modeling image and text data (e.g., Delvin et al., 2019; He et al., 2021). This work demonstrates that MAE also represents a promising approach for imputing tabular data. Compared with alternative approaches (e.g., deep generative models), MAE has the following advantages: 1) it does not require complete data during training or the knowledge about the underlying missingness mechanism; 2) without suffering the difficulties of adversarial training (Goodfellow et al., 2014) or training through variational bounds (Zhao et al., 2022), it is easier to optimize; 3) it works effectively even under a high ratio of missing data (e.g., up to 70%), which corroborates the findings in previous work (e.g. He et al, 2021) that MAE accurately reconstructs missing parts of an image, even when over half of the image is masked.
>
> To evaluate its computational cost, we further compare the running time of ReMasker (under the default setting of Table 7) and HyperImpute on datasets with over 1,000 records, with results summarized as follows (**Appendix B1**). Observe that compared with HyperImpute, ReMasker is not only more scalable but also less sensitive to data size.
>
> | | compression |  wine |  spam | credit | bike | obesity | california |
> | :----: | :----: | :----: | :----: | :----: | :----: | :----: | :----: |
> | **HyperImpute** | 154.9s | 297.7s | 185.8s | 39.9s | 259.9s | 196.1s | 195.8s |
> | **ReMasker** | 72.9s | 73.8s | 74.1s | 74.0s | 66.9s | 66.8s |66.9s |
>
> To evaluate its ability to generate new data, we run the following evaluation (**Appendix B2**). We halve each dataset into two subsets $D$ and $D'$ (both with 30% missing value missing under MAR). We train ReMasker on $D$ and then apply it to impute the missing values in $D'$. The results are shown in the table below. Note that compared with the setting where ReMasker is trained and applied on the same dataset (Figure 2), ReMasker performs comparably well under the setting that it is applied on a new dataset.
>
> | | compression |  wine |  spam | credit | bike | obesity | california |
> | :----: | :----: | :----: | :----: | :----: | :----: | :----: | :----: |
> | **RMSE** $\downarrow$ | 0.117 | 0.120 | 0.052 | 0.251 | 0.108 | 0.197 | 0.068 |
> | **WD** $\downarrow$ | 0.047 | 0.033 | 0.009 | 0.131 | 0.035 | 0.059 | 0.020 |
> | **AUROC** $\uparrow$ |  N/A | 0.784 | 0.098 | 0.921 | 0.968 | 0.937 | N/A |

---

> > ### Author Response · Authors · 2023-11-15
> >
> > > The argument presented in Section 4.1 about why Remasker generalizes different missing data assumptions could be strengthened. While transformer-based architectures have proven their superior performance in various tasks, a more comprehensive analysis of why transformers excel in missing data imputation would enhance the paper's quality and its contribution to both the missing data imputation and transformer-based research fields.
> >
> > Following the reviewer’s suggestion, we have added a detailed discussion about how ReMasker’s performance is influenced by the missingness mechanism (**Section 5 Q2**). Intuitively, ReMasker encourages learning representations invariant to re-masked values and then leverages such representations to impute missing values. Thus, while ReMasker takes effect as long as the distributions of re-masked and missing values are correlated, its performance may vary with the disparity between the two distributions. In MACR, the distributions of re-masked and missing values are highly similar; in MAR, the distribution of re-masked values is biased towards the observable values; in MNAR, there is even more disparity between the two distributions. While ReMasker works across different missingness patterns, its performance varies as MCAR > MAR > MNAR.
> >
> > Following the reviewer’s suggestion, we have added a discussion about why Transformer is effective for tabular data imputation (**Section 5 Q3**). While the theoretical understanding about the Transformer architecture is still limited, we borrow insights from recent work to explain the effectiveness of Transformer for tabular data imputation. Specifically, it is shown in (Park &
> > Kim, 2022) that the multi-head self-attention (MSA) mechanism in Transformer essentially performs (optimizable) smoothing over feature maps, which makes Transformer highly robust against severe occlusions or perturbations (Naseer et al., 2021). In the same vein, in the context of tabular data imputation, MSA benefits learning representations that are invariant to missing values.
> >
> >
> > > I'd like to understand the rationale behind using a deep encoder and a shallow decoder when describing the decoder. While the Ablation Study in Section 4.3 provides some insight, could you provide a more abstract explanation for this choice? Is there a specific reason why you believe the first fitting stage of the Remasker benefits from a more complex encoder?
> >
> > As shown in our experiments, the optimal encoder/decoder depth varies from 4 to 10. While a relatively shallow decoder often suffices to reconstruct the masked values, the encoder needs to be sufficiently complex to effectively learn the holistic representations of inputs (but not too complex to overfit). Such asymmetric encoder-decoder leads to high-fidelity imputation while minimizing the computational cost.
> >
> > > Some figures display results for different baselines (e.g., Figure 3 RMSE), and while they may appear distinct, the values on the y-axis are actually quite close. Could you elaborate on how these slight differences influence the final results? For instance, in Figure 3, Remasker outperforms other methods by only a narrow margin in terms of RMSE and AUROC. How does this marginal improvement make Remasker a more beneficial choice compared to other methods that utilize deep generative models to learn the data distribution?
> >
> > To better visualize the improvement of ReMasker over other baselines, we have summarized the performance of ReMasker and baselines (measured and ranked in terms of RMSE) on 12 benchmark datasets in Table 10 (**Appendix B3**). The key observation is that ReMasker (ranked top 1 or 2) consistently outperforms other baselines across different datasets. Also note that the only method with performance close to ReMasker is HyperImpute, which is an ensemble method that integrates multiple imputation models and automatically selects the most fitting model for each column of the given dataset. As ReMasker is a much simpler and more modular imputer, it may serve as a strong baseline for developing and evaluating future imputation methods or it can be integrated into an ensemble imputer such as HyperImpute.
> >
> >
> > Again, we thank the reviewer for the valuable feedback. Please let us know if there are any other questions or suggestions.
> >
> > Best,
> >
> > Authors

---

> > > ### Comment · Reviewer_w2Hd · 2023-11-16
> > > **Thank you for the comments**
> > >
> > > I express my gratitude to the authors for addressing my comments and conducting additional experiments to enhance the paper's quality. Their efforts not only addressed my queries but also those raised by other reviewers. I am confident that incorporating the feedback from the reviews will significantly enhance the paper's readability, particularly in terms of improving the plots and providing the necessary justifications that were lacking in the initial version of the manuscript.
> > >
> > > Upon reviewing the comments from other reviewers, it appears that the paper could benefit from a more in-depth exploration of other works utilizing transformers for tabular data, as highlighted by reviewer Fz3c. I concur with reviewer YQF tA's observation that certain results suggest a near-identical performance between Hyperimpute and the proposed method. However, I believe this similarity could be advantageous for researchers in the missing data imputation field who are familiar with alternative methods such as Deep Generative Models but have yet to encounter a paper employing transformers for this task. Although some justifications were lacking in the main manuscript, and there is room for improvement in the visual representation of results, I appreciate the authors' efforts and find the proposed method intriguing enough to merit consideration for acceptance.
> > >
> > > I will maintain my initial score but aim to collaborate with other reviewers and the AC to reach a consensus during the discussion period. Once again, I would like to thank the authors for their work and fast response to the reviews.

---

> > > > ### Author Response · Authors · 2023-11-21
> > > >
> > > > We thank the reviewer for the valuable feedback and follow-up! We are so delighted to learn that our response and additional experiments have addressed your queries. We are committed to incorporating all the comments from you and the other reviewers in our revised manuscript.
> > > >
> > > > Best,
> > > >
> > > > Authors

---

### Meta-Review · Area_Chair_MBn8 · 2023-12-09

**Metareview:**

This paper addresses the challenge of imputing missing values in tabular data and proposes a novel solution named ReMasker. To tackle this issue, ReMasker extends the masked autoencoding framework, introducing a straightforward yet highly effective approach. By randomly "re-masking" a set of values in addition to the naturally masked missing values, the auto-encoder is optimized to reconstruct this re-masked set, demonstrating competitive performance compared to state-of-the-art methods in imputation fidelity and utility across various missing-ness scenarios.

Three reviewers evaluated the paper, with two recommending acceptance and one suggesting a borderline rejection. The positive reviewers agreed that the paper is easily readable, and its ideas are simple and convincing. They also acknowledged that the experimental results are thorough and, despite their conciseness, position the proposed method as one of the best-performing options among various methods across multiple datasets and environments, making it an effective choice for practitioners. During discussions between the reviewers and the AC, strengths and weaknesses of the paper were further deliberated. Despite concerns about novelty and marginal performance improvement, all reviewers agreed that the method is a simple yet best-performing option for practitioners.

**Justification For Why Not Higher Score:**

novelty is limited. performance improvement is marginal

**Justification For Why Not Lower Score:**

I have some reservations and need to discuss with SAC on this.

---

### Decision · Program_Chairs · 2024-01-16

Accept (poster)